# Single-cell transcriptomics analysis of bullous pemphigoid unveils immune-stromal crosstalk in type 2 inflammatory disease

Tingting Liu[1,2,6], Zhenzhen Wang [ID][1,2,3,6], Xiaotong Xue[1,2], Zhe Wang [ID][1,2], Yuan Zhang[1,2], Zihao Mi[1,2], Qing Zhao[1,2], Lele Sun[1,2], Chuan Wang[1,2], Peidian Shi[1,2], Gongqi Yu[1,2], Meng Wang[1,2], Yonghu Sun [ID][1,2], Fuzhong Xue[3], Hong Liu [ID][1,2,4,7] ✉ & Furen Zhang [ID][1,2,4,5,7] ✉

Bullous pemphigoid (BP) is a type 2 inflammation- and immunity-driven skin disease, yet a comprehensive understanding of the immune landscape, particularly immune-stromal crosstalk in BP, remains elusive. Herein, using single-cell RNA sequencing (scRNA-seq) and in vitro functional analyzes, we pinpoint Th2 cells, dendritic cells (DCs), and fibroblasts as crucial cell populations. The *IL13-IL13RA1* ligand–receptor pair is identified as the most significant mediator of immune-stromal crosstalk in BP. Notably, fibroblasts and DCs expressing IL13RA1 respond to IL13-secreting Th2 cells, thereby amplifying Th2 cell-mediated cascade responses, which occurs through the specific upregulation of PLA2G2A in fibroblasts and CCL17 in myeloid cells, creating a positive feedback loop integral to immune-stromal crosstalk. Furthermore, PLA2G2A and CCL17 contribute to an increased titer of pathogenic anti-BP180-NC16A autoantibodies in BP patients. Our work provides a comprehensive insight into BP pathogenesis and shows a mechanism governing immune-stromal interactions, providing potential avenues for future therapeutic research.

Bullous pemphigoid (BP) stands as the most prevalent autoimmune sub-epidermal blistering disease, commonly affecting the elderly, especially those aged 70 years or older[1]. The global incidence of BP is approximately 0.0419 per 1000 person-years, and clinic-based prevalence averages around 0.79%. These figures can vary based on geographical location, age, gender, and the income level of a country[2]. With the global population aging, BP incidence has risen to between 2.5 and 42.8 cases per million individuals annually[3,4].

Clinically, BP is characterized by circulating IgG autoantibodies targeting epidermal basement membrane zone proteins BP180 and BP230, leading to epidermal-dermal separation, blister formation and erosions[5]. Abnormalities of cytokines from lymphocyte subsets,

including T cell-derived CXCL12[6], T helper 2 (Th2) cell-secreted cytokines IL-4 and IL-13[7–10], Tfh cell-specific cytokine IL-21[11], and Breg cell-derived TNF[12], promote B cell proliferation and the production of autoantibodies, driving the immunopathogenesis of BP. Importantly, the dominant role of Th2-mediated immunity in the pathophysiological mechanisms of BP has been widely emphasized[10,13–20]. Consequently, BP is considered one of the skin diseases predominantly associated with type 2 inflammation and immunity[8,14,21].

Type 2 inflammation, primarily driven by Th2 cells, group 2 innate lymphoid cells (ILC2), eosinophils, and type 2 cytokines, such as IL-4, IL-5, and IL-13[22,23], underlies diseases such as asthma, allergic rhinitis, and atopic dermatitis (AD)[24]. Epithelial cell-derived cytokines IL-25,

[1]Hospital for Skin Diseases, Shandong First Medical University, Jinan, Shandong, China. [2]Shandong Provincial Institute of Dermatology and Venereology, Shandong Academy of Medical Sciences, Jinan, Shandong, China. [3]Department of Biostatistics, School of Public Health, Cheeloo College of Medicine, Shandong University, Jinan, Shandong, China. [4]School of Public Health, Shandong First Medical University and Shandong Academy of Medical Sciences, Jinan, Shandong, China. [5]Shandong University of Traditional Chinese Medicine, Jinan, Shandong, China. [6]These authors contributed equally: Tingting Liu, Zhenzhen Wang. [7]These authors jointly supervised this work: Hong Liu, Furen Zhang. ✉e-mail: hongyue2519@hotmail.com; zhangfuren@hotmail.com

IL-33, and thymic stromal lymphopoietin are crucial in activating Th2 cells and fostering type 2 inflammatory responses[8,25,26]. This highlights the pivotal role of epithelial cells and keratinocytes in Th2-mediated immunity, implicating stromal cells in the development of type 2 inflammatory diseases. However, a comprehensive understanding of immune-stromal crosstalk and mechanisms governing Th2-mediated immunity in type 2 inflammation remains elusive.

Furthermore, when comparing the two pivotal cytokines of type 2 inflammation, IL-4 appears to be more relevant for central aspects of the Th2 response within lymph nodes, while IL-13 may play a more prominent role in peripheral tissues[27]. This suggests that IL-13 might be a more important mediator for Th2 immunity in skin-related conditions[28]. For instance, IL-13 is the dominant type 2 cytokine in lesions from patients with AD and BP[18,29,30]. However, the specific mechanisms underlying the interactions between IL-13-producing cells and IL-13-responsive cells are not yet fully understood. Moreover, despite patients with BP and AD sharing common clinical and pathological features and Th2 cell-mediated immunity, pathogenic autoantibodies are exclusively present in BP. Unlike AD, where dupilumab monotherapy (a monoclonal antibody that blocks IL-4 and IL-13 signaling) significantly improves disease activity and reverses AD-associated epidermal abnormalities[31], dupilumab primarily reduces the dosage of corticosteroids in BP[32]. This demonstrates the complex and distinct pathophysiological mechanisms underlying BP, warranting further research for better understanding.

In this work, to comprehensively investigate IL-13-related responses in Th2-mediated immunity and explore the intricate interplay between immune and stromal cells, we employ single-cell RNA sequencing (scRNA-seq), cell-cell communication analysis and in vitro functional analysis on various sample types from BP patients and controls. Our findings advance the understanding of immune-fibroblast communication and emphasize the role of the *IL13-IL13RA1* axis in Th2-mediated immunity in BP patients, with potential implications for broader research in autoimmune diseases.

## Results

### Cellular composition of BP lesions

To explore the cellular and molecular mechanisms that orchestrate the established Th2-mediated immunity in BP, we performed single-cell RNA sequencing (scRNA-seq) from five lesions of BP patients and eight normal skin of healthy donors during the discovery stage (Fig. 1a, Supplementary Table 1). After quality control filters (Supplementary Fig. 1a), a total of 68,374 skin cells were included for further scRNA-seq analysis and 20 clusters were identified using a graph-based method (Supplementary Fig. 1b, c, Supplementary data 1). By overlapping the cluster marker genes with manual curation of canonical markers, nine main cell types: keratinocytes (KC; *KRT1*, *KRT5*, *KRT10*, *KRT14*), fibroblasts (*DCN*, *COL1A1*, *COL1A2*), dendritic cells/macrophages (DC/Mac; *PTPRC*, *CD68*, *CD1C*), T/Nature killer (NK) cells (*PTPRC*, *CD3D*, *GNLY*, *NKG7*), endothelial cells (*CD93*, *ACKR1*, *AQP1*), lymphatic cells (*CCL21*, *LYVE1*, *TFF3*), melanocytes (*TYRP1*, *PMEL*, *DCT*), sweat gland cells (*DCD*, *KRT19*, *AQP5*) and smooth muscle cells (*TAGLN*, *ACTA2*, *TPM2*) were identified (Supplementary Fig. 1c and Supplementary data 2). All of these cell subtypes were shared by BP patients and healthy controls (Supplementary Fig. 1d–f). Cell composition analysis revealed an increase in the proportion of T/NK and DC/Mac cells in lesional skin relative to healthy control (Supplementary Fig. 1g). The increase of CD3[+] T cells and CD68[+] macrophages within the lesion of BP patients were confirmed by immunochemistry staining (Supplementary Fig. 1h).

### BP lesions are characterized by a type 2 immune environment

Given that demarcating immune cell populations might reveal new cell type-specific expression differences, we sub-clustered all immune cells of T/NK and DC/Mac clusters (Cluster 12 and 13 in Supplementary

Fig. 1b). The sub-clustering of all immune cells identified 14 sub-clusters, including six major sub-clusters of T/NK cells and eight major sub-clusters of myeloid cells (Fig. 1b, c, Supplementary data 3). On the basis of the expression profile, the six T/NK subpopulations were Memory CD4 T, Effector CD8 T, NK and gdT cells, Treg, Th2 and Proliferating T cells; the eight major sub-clusters of myeloid cells were *LAMP3*[+] DC, cDC2, *CXCL1*[high] DC, *CD163*[+] Macro, *FCN1*[+] Macro, *GPNMB*[high] Macro, Langerhans cells, pDC and Mast cells (Supplementary Fig. 2a–e, Supplementary data 4). The canonical markers utilized for annotating each cluster are presented in Supplementary Fig. 2c, e.

In the six T/NK subpopulations, levels of Th2 (*GATA3*[+]*IL13*[+]) T cells were expanded remarkably in lesional samples in patients with BP (Fig. 1d, Supplementary Fig. 2a, b). Moreover, an increased trend was observed in Proliferating T subset (*MKI67*[+]*STMN1*[+]) in BP lesions, which expressed high level of Th2-related marker genes, including *IL13* and *IL5* (Fig. 1d, Supplementary Fig. 2c). The proportional analysis of the eight myeloid subpopulations showed that Langerhans cells were reduced in skin from BP patients. Conversely, the cDC2 cluster with high expression of the type 2 chemokines *CCL17* and *CCL22*[33], increased in skin from patients compared to skin from healthy donors. Similarly, *LAMP3*[+] DC and *CXCL1*[high] DC, both exhibiting high expression *CCL17* and *CCL22*, also showed an increased trend in patients (Fig. 1e, Supplementary Fig. 2a, d). The M2 type macrophage, *CD163*[+] Macro cluster with high amounts of *CD163*, *CCL18* and *CCL13*, showed an increased trend in lesional BP versus controls (Fig. 1e, Supplementary Fig. 2e). All of those confirmed that BP lesions were characterized by a type 2 immune environment, with the dramatic enrichment of Th2 cells and type 2 chemokine-expressing DC cells, and the increase trend of M2 macrophage (*CD163*[+] Macro) and type 2 cytokine-expressing Proliferating T cells. Immunofluorescence co-staining also validated the increase of Th2 cells by the relative expansion of CD3[+]GATA3[+] cells (Fig. 1f) and CD3[+]IL-13[+] cells (Supplementary Fig. 2f) within the lesion of BP patients, immunochemistry staining confirmed the infiltration of CD11c[+] DC cells, mainly beneath or near the bullae in the dermis (Fig. 1g). Due to the marked elevation of DCs, the proportion of Langerhans cells among myeloid cells in BP decreased compared to normal controls. However, an increase in the number of Langerhans cells was observed via immunohistochemistry staining in BP patients (Supplementary Fig. 2g), which aligns with findings from previous publications[34], indicating the significance of Langerhans cells in BP pathogenesis. Taken together, these data suggested that patients with BP are characterized by a type 2 immune environment, both in lymphocytes and myeloid cells.

### Compositional differences in fibroblasts and keratinocytes

Cutaneous fibroblasts (FBs) provide a mechanically resilient, adhesive yet elastic structural foundation[35], which are found to be heterogeneous and participate in multiple skin conditions including atopic dermatitis (AD)[33], vitiligo[36] and psoriasis[37,38]. Then we explored FBs heterogeneity in BP patients and 10 subgroups on second-level analysis were identified (Fig. 2a–c, Supplementary Fig. 3a, b, Supplementary data 5). The largest FB population cluster 0 (*CCL19*[+] FB) expressed *APOE* and *C3*, along with inflammatory chemokines *CCL19* and *CCL2*, which is a pro-inflammatory FBs and was reported to regulate the recruitment and organization of lymphocytes and myeloid cells[33,39]. Cluster 1 (*APCDD1*[+] FB) characterized by expression of *APCDD1* and *WIF1*, cluster 4 (*POSTN*[+] FB) and 5 (*ASPN*[+] FB) with *POSTN* or *ASPN*[39], were identified as secretory-papillary FBs and mesenchymal FBs[39], respectively. Cluster 2 (*CFH*[+] FB) expressed the complement component genes (e.g., *CFD*, *CFH*, and *C3*), cluster 3 (*HSPA1A*[+] FB) upregulated expression of anti-apoptotic genes *HSPA1A* and *HSPA1B*[40], cluster 6 (*FBLN1*[+] FB) was found to highly express *FBLN1*, *MFAP5* and *ANGPTL1*[33]. The last three clusters with low number of cells were annotated by the specific biomarker of *IGFBP2* and *FGFBP2* in cluster 7 (*IGFBP2*[+] FB)[41], *INHBA* in cluster 8 (*INHBA*[+] FB)[42], *TAGLN* in cluster 9 (*TAGLN*[+] FB)[43]

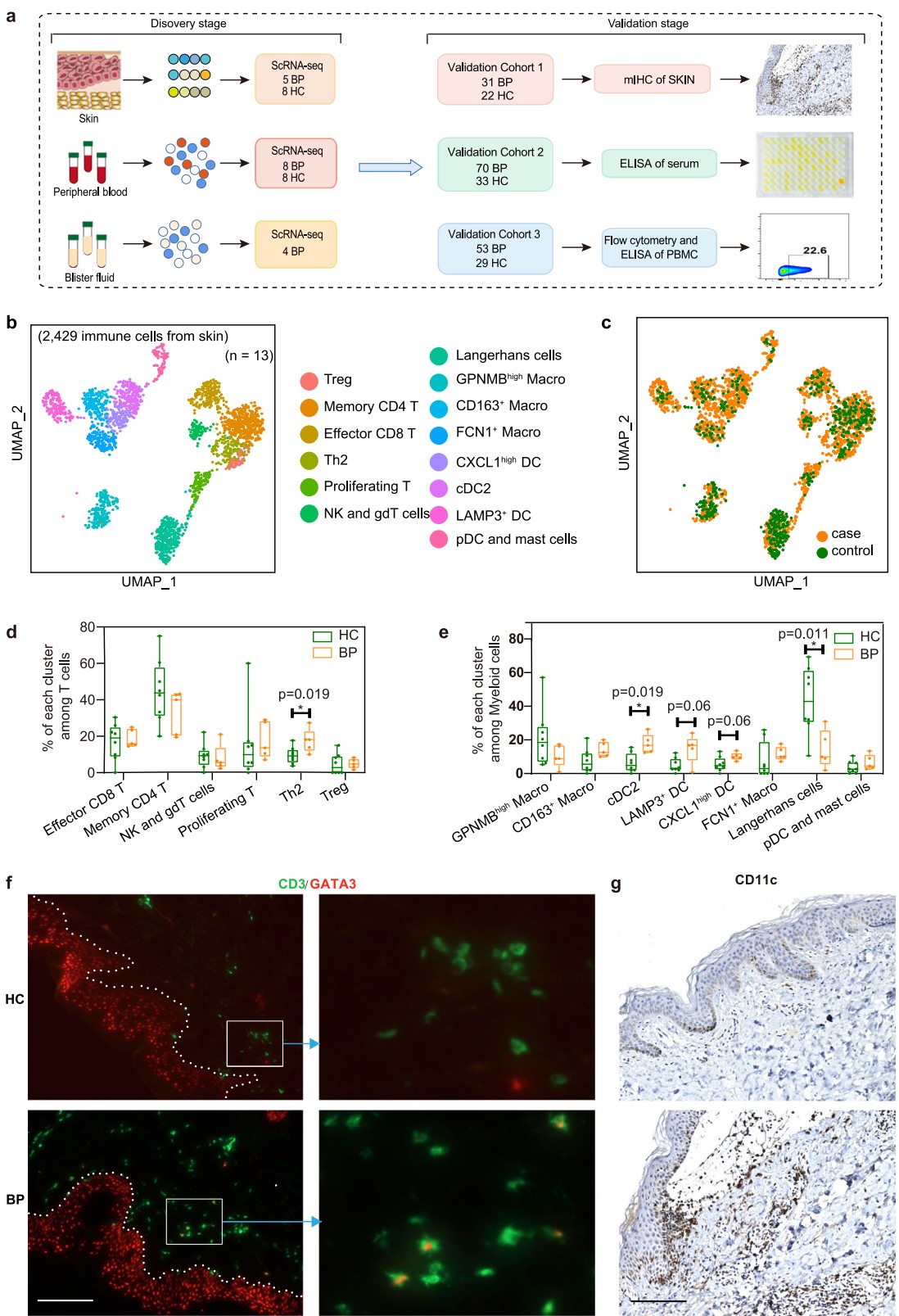

(Supplementary Fig. 3b, Supplementary data 6). The diversity and complexity of these clusters revealed a remarkable degree of functional heterogeneity within skin fibroblasts, reflecting a wide range of potential functions in inflammation, immune responses, metabolism and adipogenesis. Further, the cell composition analysis revealed that cluster 1 (*APCDD1*+ FB) FBs were more abundant in lesional BP samples (Fig. 2d).

Keratinocytes (KCs) constitute the majority of cells sequenced (49,132 cells), where the circulating autoantibodies targeted structural proteins BP180 and BP230 were located. Although we did not observe proportional changes between lesional BP versus controls in overall KCs (Supplementary Fig. 1g), after sub-clustering analysis, we found significant proportional differences in five of the 12 sub-clusters (Fig. 2e–h, Supplementary Fig. 3c, d, Supplementary data 7). The 12

**Fig. 1 | Sub-clustering of immune cells from the skin scRNA-seq dataset.**
**a** Flowchart depicting the overall experimental design of this study. The number of patients with bullous pemphigoid (BP) and healthy controls (HC) for the discovery cohort and each validation cohort were indicated. **b** UMAP visualization of the sub-clustering of skin immune cells of the discovery cohort. **c** UMAP plot for skin immune cells split by BP patients and HC. Cells from patients with BP are shown in orange, whereas cells from HC are shown in green. **d** Percentage of each T cell subset among all T cells between BP patients ($n = 5$) and HC ($n = 8$). **e** Frequency of each myeloid subset among all myeloid cells between BP patients ($n = 5$) and HC

($n = 8$). *P*-values in **d** and **e** were calculated using two-sided Mann–Whitney U-test. Each sample is represented as one dot. *$P < 0.05$. In the box plot **d** and **e**: Minima: Lower limit of the whisker. Maxima: Upper limit of the whisker. Center: Median line inside the box. The upper and lower box bounds represent the 25% and 75% percentile of data. **f** Immunofluorescence co-staining showing the co-localization of GATA3 and T cell marker CD3 in BP lesions. scale bar = 150 μm. **g** Immunochemistry staining showing the CD11c-expressed DC cells within the lesions of BP patients. scale bar = 150 μm. Data are from three independent experiments in **f** and **g**. DC, Dendritic Cells.

sub-clusters can be identified as seven KC states on the basis of previously reported expression profile[33] (Supplementary Fig. 3d). Stratum corneum KCs highly expressed *ASPRV1*, *LOR*, *FLG* and *FLG2*, including clusters 9 and 10. Suprabasal KCs specifically expressed early differentiation keratins (*KRT1* and *KRT10*), which were consisted of cluster 0, cluster 2 and cluster 7. The outer root sheath cells are corresponded to the cluster 5, and the inner root sheath sebaceous (cluster 8) expressed *APOC1*. Basal KCs highly expressed *KRT5*, *KRT14*, and *KRT15*, specifically separating into three clusters 1, 6 and 11. A cluster (cluster 4) regulating ion channels specifically expressed "Channel-ATPase" genes (*ATP1B1*, *ATP1A1* and *SAT1*) and "channel-gap" genes (*GJB6*). The proliferating KCs (cluster 3) were characterized by expression of *MKI67*, *STMN1*, *UBE2C* and *TOP2A* (Supplementary Fig. 3d, Supplementary data 8). 0_suprabasal, 6_basal KCs were predominated by healthy control (HC) samples, whereas 7_suprabasal, 5_outer root sheath and 3_proliferating KCs overwhelmingly comprised lesional BP cells (Fig. 2h). These results suggest that the epidermal compartments are differentially distributed in the skin from BP patients and HC.

### *IL13-IL13RA1* signaling: key mediator of Th2 cell communication with IL-13-responsive fibroblasts and myeloid Cells in BP patients

Considering CellChat is an effort tool that is able to quantitatively infer and analyze intracellular communication networks from scRNA-seq data, we ran CellChat analysis on 22 clusters including all the 14 immune populations, the five KC and one FB populations differentially enriched in cases and controls, 1_Basal KCs which express the structural protein BP180 encoded gene (*COL17A1*), and *CCL19*+ FB that implicated in the pathogenesis of AD[33], to predict major signaling inputs and outputs.

By comparing the overall communication probability between normal and lesional skin, we found that 15 signaling pathways were exclusively enriched in BP lesional skin, including SPP1, IL4, CD23, TRAIL, IL10, CD226, SN, IL2, CD80, IL6, AGRN, KIT, CNTN, SEMA5 and CADM (Fig. 3a), suggesting that these pathways might critically contribute to disease progression. Specific to IL4 signaling, the canonical Th2-related signaling, network centrality analysis identified that Th2 cells are the most prominent sources in BP patients (Fig. 3b). Given that IL4 signaling contains *IL4-*, *IL5-* and *IL13-* related ligand–receptor pairs, we further examined the expression and secretion of these cytokines in BP patients, founding that total T cells and Th2 cells expressed significantly higher amounts of *IL4*, *IL5* and *IL13* (Fig. 3c), and patients secreted increased levels of IL-4, IL-5 and IL-13 (Fig. 3d). Noteworthy among the three Th2 cytokines, the expression of IL-13 was much higher than IL-4 and IL-5 both at mRNA and protein levels.

Consistently, the *IL13-IL13RA1* was identified as the most significant ligand–receptor pair in IL4 signaling (Fig. 3e), contributing to the communication from Th2 cells to six different clusters, including two fibroblast clusters (*CCL19*+ FB and *APCDD1*+ FB) and four myeloid clusters (*LAMP3*+ DC, cDC2, *GPNMB*high Macro and Langerhans cells) (Fig. 3f). Taken together, the predominance of *IL13* expression and the significance of *IL13-IL13RA1* pair indicated the critical role of *IL13* in the pathogenesis of BP.

### In IL-13-responsive fibroblasts: PLA2G2A upregulation drives immune cell aggregation via the CXCL12/CXCR4 axis in lesional BP sites

To further explore the contribution of IL-13-responsive fibroblast cells to the disease, we firstly performed a differential gene expression analysis in the two IL-13-responsive fibroblast clusters, *CCL19*+ FB and *APCDD1*+ FB. The analysis demonstrated that genes with type 2 inflammation (*POSTN* and *TNC*), the inflammatory cytokines and chemokines (*CCL26* and *CCL19*) were upregulated in lesional BP versus controls (Fig. 4a, b, Supplementary data 9 and 10), corresponding to the findings in AD patients[33]. Intriguingly, *PLA2G2A*, encoding secretory calcium-dependent phospholipase A2, which was reported to recruit immune cells in breast cancer[44], was highly expressed in both *CCL19*+ FB and *APCDD1*+ FB from BP patients (Fig. 4a, b, Supplementary data 9 and 10). Then *PLA2G2A* was found to be exclusively expressed in fibroblasts among all skin cell types, and an increased trend was observed in total fibroblasts from lesional BP patients, despite without statistical significance (Fig. 4c). Immunofluorescence co-staining of the fibroblast marker PDGFRA alongside PLA2G2A revealed an elevation of PLA2G2A within fibroblasts in BP lesions (Supplementary Fig. 4a). Moreover, we validated significant upregulation of secreted PLA2G2A in the serum from BP patients by ELISA (Fig. 4d). Furthermore, a positive correlation was observed between serum PLA2G2A and both BSA scores and the maximum dosage of corticosteroids in BP patients, respectively (Supplementary Fig. 4b, c). These results suggest that serum PLA2G2A could serve as a promising marker for assessing the disease severity. To further explore the role of *PLA2G2A*, we treated THP1 and Jurkat cells with PLA2G2A recombinant protein and found that PLA2G2A promoted the migration of the two cells (Fig. 4e), indicating the potential of *CCL19*+ FB and *APCDD1*+ FB to attract immune cells by overexpression of *PLA2G2A*. Immunofluorescence analysis also showed a similar spatial distribution that PLA2G2A+ fibroblasts situated between CD68+ macrophages and CD3+ T cells (Fig. 4f). Consistently, GO pathway analysis showed lesional *CCL19*+ FB and *APCDD1*+ FB with enrichment of terms such as positive regulation of leukocyte chemotaxis, positive regulation of leukocyte migration, regulation of cell migration, cell adhesion and biological adhesion (Supplementary Fig. 4d, e).

To further explore the underlying mechanism involved in the PLA2G2A-mediated immune infiltration, CellChat was used to further analyze increased communication signals among these two IL-13 responsive fibroblast clusters and other clusters. The *MIF-CD74* pair emerged as the most robust signal across all cell clusters, while *CXCL12-CXCR4* pair, the second most significant, showed notable enrichment specifically within fibroblast subpopulations (Fig. 4g, Supplementary Fig. 5). Then the *CXCL12-CXCR4* ligand–receptor pair originating from both FB0 and FB1 was selected (Supplementary Fig. 5), which has been reported to play a pathogenic role in modulating B cell trafficking and differentiation in BP patients[6]. Compared to healthy controls, *CXCL12-CXCR4* pair was also found to be highly active in lesions, which acted in paracrine manner from these two fibroblast clusters to nearly all immune clusters (Fig. 4g, h, Supplementary Fig. 5). According to the expression profiles, we then found *CXCL12* was expressed in fibroblasts, smooth muscle cells and

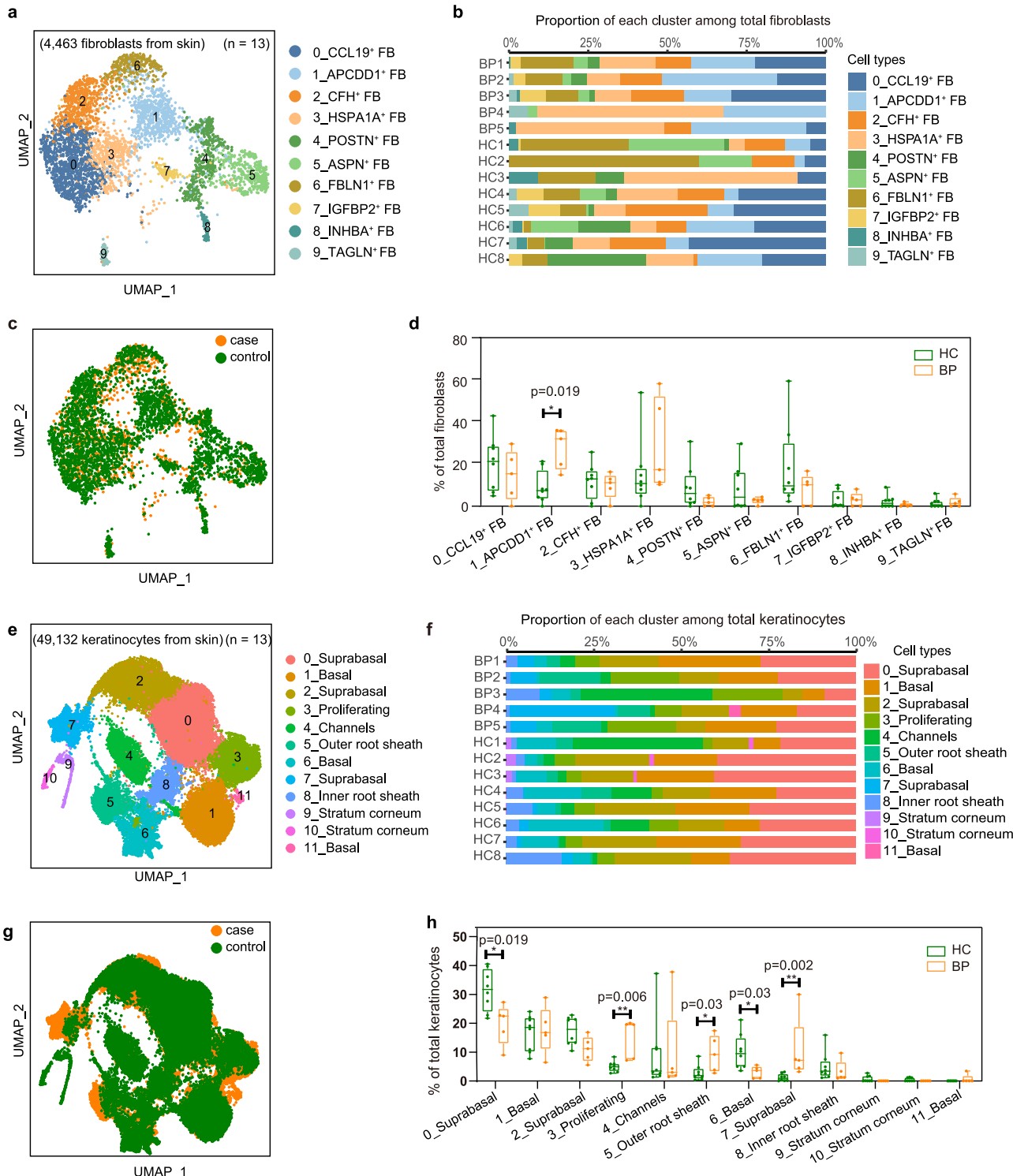

**Fig. 2 | Heterogeneity of fibroblasts and keratinocytes from the skin scRNA-seq dataset. a** UMAP plot of the fibroblasts (FB) colored by major cell clusters. **b** Bar plots illustrate the relative contributions of fibroblasts cell types in each sample. **c** UMAP plot for fibroblasts split by BP patients and HC. Cells from patients with BP are shown in orange, whereas cells from HC are shown in green. **d** Percentage of each fibroblast subset among all fibroblasts between BP ($n = 5$) and HC ($n = 8$). **e** UMAP visualization of the sub-clustering of keratinocytes of the discovery cohort. **f** Bar plots illustrate the relative contributions of keratinocytes cell types in each

sample. **g** UMAP plot for keratinocytes split by BP patients and HC. **h** Frequency of each keratinocyte subset among all keratinocytes between BP ($n = 5$) and HC ($n = 8$). *P*-values in **d** and **h** were calculated using two-sided Mann–Whitney U-test, only *P*-values < 0.05 are shown. In the box plot **d** and **h**: Minima: Lower limit of the whisker. Maxima: Upper limit of the whisker. Center: Median line inside the box. The upper and lower box bounds represent the 25% and 75% percentile of data. Each sample is represented as one dot. *$P < 0.05$, **$P < 0.01$.

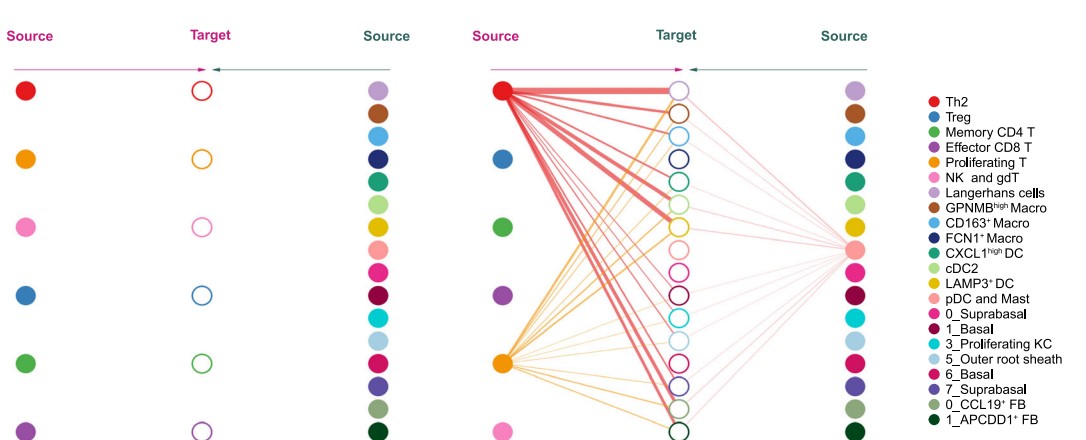

**a** (BP vs Control: Relative information flow)

**b** IL4 signaling pathway network in BP patients

**c** Total T cells / Th2 mRNA expression (HC vs BP)

**d** serum IL-4, IL-5, IL-13 by ELISA (pg/ML) (HC vs BP)

**e** IL13 – IL13RA1 and IL13 – (IL4R+IL13RA1) communication probability

**f** IL13 – IL13RA1 signaling network in BP patients

endothelial cells, but there was no dramatic difference between controls and cases among these clusters (Supplementary Fig. 6a). Although the *CXCL12* didn't change dramatically, its receptor CXCR4 was exclusively expressed by immune cell clusters, and a trend toward increased expression of *CXCR4* was observed in the T/NK cells in lesional skin relative to healthy control (Supplementary Fig. 6b). The immunofluorescence confirmed the expression of CXCL12 in

fibroblasts and CXCR4 in immune cells within BP lesions (Supplementary Fig. 6c). These results implied that the *CXCL12-CXCR4* ligand–receptor may participate in the immune infiltration of T/NK cells.

Considering the shared immune recruitment function of PLA2G2A and CXCL12/CXCR4, we then explored the correlation of expression level of PLA2G2A and CXCL12 in fibroblasts. Significant positive

**Fig. 3 | The *IL13-IL13RA1* mediates the interaction between Th2 and fibroblasts/myeloid cells. a** Significant signaling pathways were ranked based on differences in the overall information flow within the inferred networks between BP and HC skin scRNA-seq dataset. We have added the abbreviations in Supplementary data 15. **b** Circle plot of the inferred IL4 signaling pathway among major cell types in the BP group. **c** Bar charts showing *IL4*, *IL5* and *IL13* mRNA expression within total T cells (left) and Th2 subpopulation (right) between BP (*n* = 5) and HC (*n* = 8). **d** Protein levels of IL-4, IL-5 and IL-13 in serum from BP (*n* = 43) and HC (*n* = 29) examined by ELISA. In the box plot **c** and **d**: Minima: Lower limit of the whisker. Maxima: Upper limit of the whisker. Center: Median line inside the box. The upper and lower box bounds represent the 25% and 75% percentile of data. Each sample is represented as one dot. **e** Comparison of the multiple ligand–receptor pairs among IL4 signaling pathway originated from Th2 cells. *P*-values were computed from one-sided permutation test (CellChat 1.5.0). **f** Hierarchical plot showing inferred intercellular communication network of *IL13-IL13RA1* signaling in BP skin. Left and right portions show autocrine and paracrine signaling, respectively. In the left part: The six T cell clusters in the middle acted as the signal-receiving cells. The left half of the diagram depicts autocrine signal, which refers to IL13 signal released by the six T cell clusters that act on themselves. Correspondingly, the right half shows paracrine signal, which IL13 signal released by other types of cells that act on the six T cell clusters. In the right part: The non-T cell clusters acted as the signal-receiving cells. The left half of the diagram depicts paracrine signal, which IL13 released by six T cell clusters that act on these non-T cell clusters. While, the right half shows autocrine signals, which are signals released by these non-T cell clusters that act on themselves. *P*-values in **c** and **d** were calculated using two-sided Mann–Whitney U-test. **P* < 0.05, ***P* < 0.01, *****P* < 0.0001.

correlation between the expression of *PLA2G2A* and *CXCL12* was observed in all fibroblasts. Due to the upregulation of *PLA2G2A*, the correlation was much stronger in BP lesions, compared to normal samples (Fig. 4i). Immunofluorescence confirmed the co-expression of CXCL12 and PLA2G2A in BP lesions (Supplementary Fig. 6d). Then we treated peripheral blood mononuclear cell (PBMC) with PLA2G2A recombinant protein and found that PLA2G2A increased the expression of CXCR4 on CD3⁺ T cells (Fig. 4j), indicating the potential role of PLA2G2A to recruit T cells via CXCL12/CXCR4 axis. Collectively, these results implied that the overexpression of *PLA2G2A* from fibroblasts promoted immune infiltration through CXCL12/CXCR4 axis.

### *CCL17* is increased in IL-13-responsive myeloid cells in BP patients

After the infiltration of immune cells, to investigate the cellular and molecular mechanisms that orchestrate the type 2 immune environment in lesions from BP patients (Fig. 1), the interacting pairs specially targeted Th2 and type 2 cytokine-expressing Proliferating T cells were analyzed. The significant mean top 50 interacting pair by CellPhoneBD identified that *CCL17-CCR4* signaling was specifically derived from IL-13-responsive DC clusters to Th2 and type 2 cytokine-expressing Proliferating T cells (Supplementary Fig. 7). CellChat also confirmed the *CCL17-CCR4* ligand–receptor pair, contributing to the communication from *LAMP3⁺* DC, cDC2 and *CXCL1*^high DC cells to Th2 and proliferating T cells (Fig. 5a).

According to the expression profiles, we found that *CCL17* was exclusively produced by DC/Mac cells (Fig. 5b), and further recluster analysis confirmed that *CCL17* was mainly produced by *LAMP3⁺* DC cluster, and *CCL17* was increased in myeloid clusters of *GPNMB*^high Macro, *CD163⁺* Macro and *CXCL1*^high DC cells from BP lesions (Supplementary Fig. 8a). We also validated significant upregulation of CCL17 secretion in the serum from BP patients by ELISA (Fig. 5c). *CCR4*, the receptor of *CCL17*, was exclusively expressed by T cells (Fig. 5d), but we didn't see dramatic changes of *CCR4* expression in total T cells and Th2 cells (Fig. 5d, Supplementary Fig. 8b). Correspondingly, the recombinant protein of CCL17 treatment didn't affect the expression of CCR4 on CD3⁺ T cells from PBMCs (Fig. 5e). Considering the comparable expression of CCR4 and the elevated production of CCL17, we hypothesized that the CCL17 was involved in the orchestration of Th2-mediated immunity, rather than the recruitment of T cells by CCL17/CCR4 axis.

### Fibroblast-derived PLA2G2A drives the secretion of CCL17

Although the relationship between CCL17 levels and clinical manifestations has been reported[45–47], the mechanisms underlying the upregulation of CCL17 in skin conditions, including BP, have remained largely understood. Given both DCs and fibroblasts are the main cell types that respond to IL-13 (Fig. 3f), and the increased expression of PLA2G2A in fibroblasts attracts the infiltration of immune cells including DCs through CXCL12-CXCR4 (Fig. 4h), we then performed correlation analysis of the two key factors, PLA2G2A and CCL17, derived from fibroblasts and myeloid cells respectively. A significant positive correlation between the secretion of PLA2G2A and CCL17 in serum was observed (Fig. 5f), and treatment of PLA2G2A recombinant protein stimulated the secretion of CCL17 from PBMCs both in cases and controls (Fig. 5g). To explore whether CCL17 was secreted by DCs, DCs sorted from both BP patients and controls were stimulated with PLA2G2A in vitro, resulting in elevated secretion of CCL17 after stimulation (Supplementary Fig. 8c). These results indicated that the fibroblast-derived PLA2G2A may drive the secretion of CCL17 from myeloid cells, especially the DC clusters, in BP patients.

### PLA2G2A and CCL17 play critical roles in IL-13 secretion

Given the *CCL17-CCR4* signaling specifically targeted Th2 cells and type 2 cytokine-expressing Proliferating T cells, we then tested the role of CCL17 in the Th2 (IL-13)-mediated immunity in BP patients. Firstly, treatment of CCL17 recombinant protein induced a pronounced increase in the secretion of IL-13 from PBMCs in BP cases (Fig. 5h). Consistently, T cells sorted from BP patients were stimulated with CCL17 in vitro, leading to increased secretion of IL-13 after stimulation (Supplementary Fig. 8d). As we found PLA2G2A stimulated CCL17 production and CCL17 induced IL-13 secretion, we then checked if the PLA2G2A could upregulate IL-13, and found that IL-13 release was elevated dramatically from PBMCs and sorted T cells stimulated with PLA2G2A treatment in vitro (Fig. 5i, Supplementary Fig. 8e).

Overall, fibroblast-derived PLA2G2A drove secretion of myeloid-derived CCL17, which further induced the IL-13-mediated immunity in the development of BP and type 2 inflammatory diseases, suggesting that PLA2G2A and CCL17 serve as promising therapeutic targets in BP.

### Both PLA2G2A and CCL17 promote anti-BP180-NC16A antibody production

In BP, autoantibodies directed against hemidesmosome proteins BP180 and/or BP230 promote the activation of complement, degranulation of mast cells, and activation of neutrophils, leading to the development of sub-epidermal blisters[48]. Therefore, we explored the effects of the two promising therapeutic targets, PLA2G2A and CCL17, on the secretion of anti-BP180 and anti-BP230 antibodies. The anti-BP180-NC16A antibody titers were increased markedly in supernatants of CCL17 stimulated patient-derived PBMCs, but not in the supernatants from healthy controls (Fig. 5j). Consistently, PLA2G2A stimulation triggered BP180-NC16A autoantibodies production in PBMCs from patients with BP, not from the healthy controls (Fig. 5k). Although we tested the anti-BP230 antibody simultaneously, positive response wasn't observed in either CCL17- or PLA2G2A-treated PBMCs both from patients and healthy controls (Supplementary Fig. 8f, g).

To address the underlying mechanism involved in the secretion of anti-BP180 antibody, PBMCs were thus stimulated with PLA2G2A or CCL17 to test whether they mediated B cell differentiation in BP.

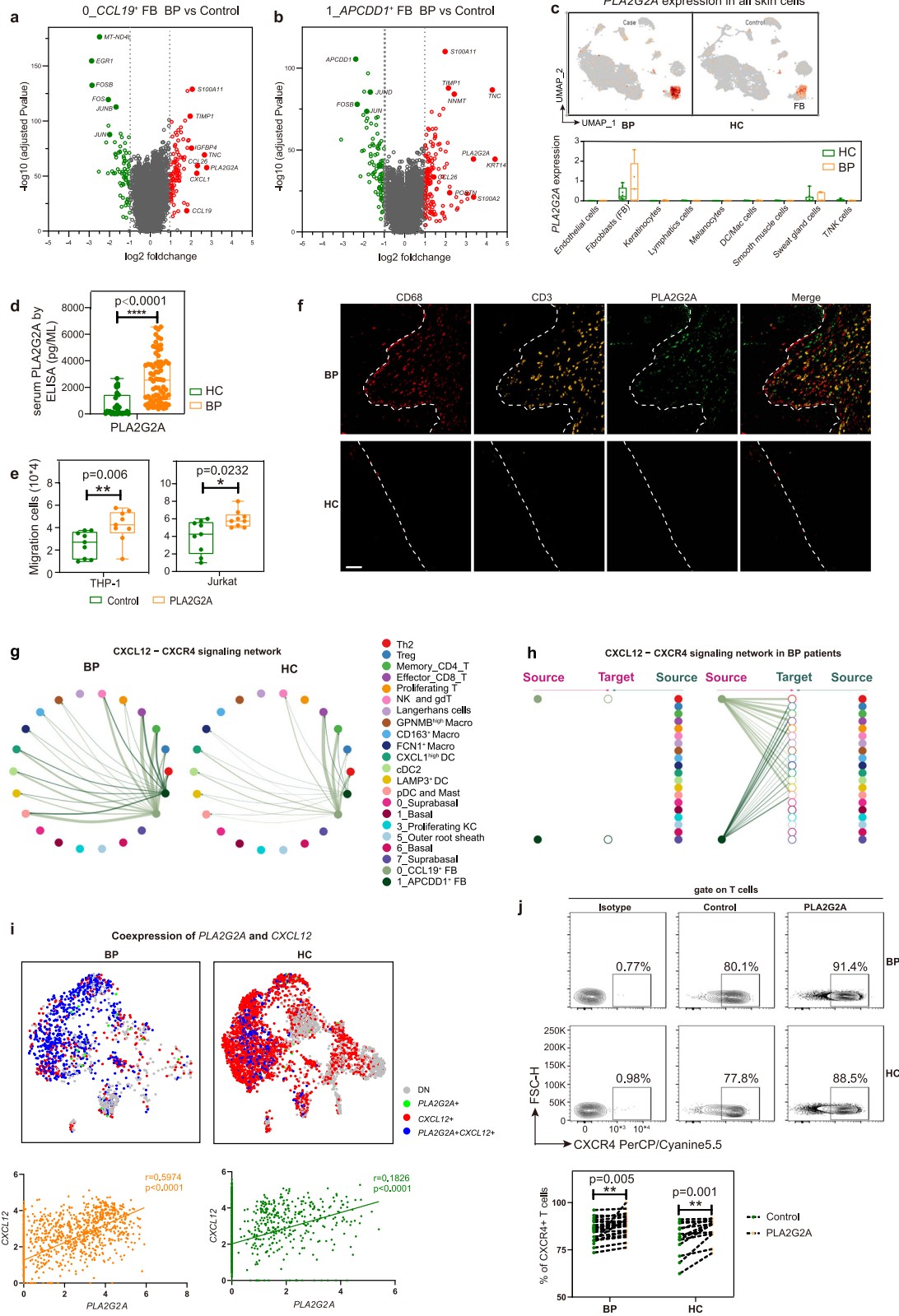

The flow cytometry results showed that both PLA2G2A and CCL17 induced a significant increase in the proportion of CD138+ plasma cells of total B cells (Supplementary Fig. 9) in BP patients.

Collectively, fibroblast-derived PLA2G2A and DC-derived CCL17 stimulated the secretion of anti-BP180-NC16A antibody, further elucidating the immunologic pathogenesis of BP, and confirming CCL17 and PLA2G2A are candidates for therapeutic strategy.

**The distorts of cell-cell communication mediated by *IL13-IL13RA1* are markedly predominant in PBMC and blister samples**

To provide additional support for the crosstalk between different subsets observed in lesions from BP patients, we performed scRNA-seq for the PBMCs of eight BP patients and eight HC and blister of four BP patients (Fig. 1a, Supplementary Table 1). In total, after quality control, 102,483 cells (Supplementary Fig. 10a) and 19,722 cells

**Fig. 4 | The immune-stromal cell crosstalk recruits immune cells via CXCL12/CXCR4 axis mediated by PLA2G2A in BP patients. a** Volcano plot of gene features of the *CCL19⁺* FB cluster in BP patients compared to HC. **b** Volcano plot of gene features of the *APCDD1⁺* FB cluster in BP patients compared to HC. *P*-values in **a** and **b** were obtained using the two-sided Likelihood-ratio test and Bonferroni corrected (Seurat 4). **c** The feature plot and the bar chart showing *PLA2G2A* mRNA expression within total skin cells between BP (*n* = 5) and HC (*n* = 8). **d** Protein level of PLA2G2A in serum from BP (*n* = 73) and HC (*n* = 31) examined by ELISA. **e** Transwell assays were used to measure cell migration of THP1 (*n* = 9) and Jurkat (*n* = 9) cells treated by PLA2G2A recombinant protein. *P*-values in **c**–**e** were calculated using two-sided Mann–Whitney U-test. **P* < 0.05, ***P* < 0.01, *****P* < 0.0001. In the box plot **c**–**e**: Minima: Lower limit of the whisker. Maxima: Upper limit of the whisker. Center: Median line inside the box. The upper and lower box bounds represent the 25% and 75% percentile of data. **f** Representative images of a BP patient and HC stained by multicolored IHC; green represents PLA2G2A⁺ fibroblasts, red represents macrophages, and yellow represents CD3 T cells. scale bar = 50 μm. Data are from three independent experiments. **g** Circle plots of the inferred *CXCL12-CXCR4* pathway among major cell types in the BP and HC groups. **h** Hierarchical plot showing inferred intercellular communication network of *CXCL12-CXCR4* signaling in BP skin. **i** The *PLA2G2A/CXCL12* gene pair co-expression (upper panel), and the gene expression correlation analysis in expressing both genes (lower panel) in fibroblasts. The correlation was measured using the Pearson correlation coefficient. *P*-values were calculated using two-sided Pearson correlation test. **j** Flow plots of CD3⁺ T cells from PBMCs showing the expression of CXCR4 treated by PLA2G2A recombinant protein (upper panel), and the frequency of CXCR4 (lower panel) in BP (*n* = 21) and HC (*n* = 13) groups. *P*-values were calculated using paired two-sided Student's *t*-test. ***P* < 0.01. Each sample is represented as one dot.

(Supplementary Fig. 10b) were obtained from PBMC and blister scRNA-seq data, respectively.

In the PBMC scRNA-seq dataset, after a series of cluster and recluster analysis (Supplementary Fig. 10c–f and 11), 21 cell types were identified, including the main disease-associated cell types Th2, Proliferating T and DCs (Fig. 6a, Supplementary data 11 and 12). By the CellChat analysis, IL4 signaling was enriched (Fig. 6b) and several ligand–receptor pairs (Fig. 6c) were identified. Among multiple ligand–receptor pairs in IL4 signaling, *IL13-IL13RA1* was found to act as the major ligand–receptor pair, which was activated in BP patients, orchestrating the crosstalk between Th2/Proliferating T cells and myeloid cells, especially the DC clusters (Fig. 6c, Supplementary Fig. 12a, b), which was paralleled with the communication observed in leisional samples.

Correspondingly with the PBMC and lesional samples, the Th2/Tc2 (cluster 6; *CD3D, LMO4, GATA3, NBAS, IL17RB*) and *IL13/IL5*-expressing Proliferating T (cluster 12; *CD3D, TYMS, MKI67, TOP2A*) cells were also observed in the four blister samples (Fig. 6d, e, Supplementary Fig. 12c, d, Supplementary data 13 and 14). Based on the gene signatures reported by Schulte-Schrepping and colleagues[49], cluster 7 was identified as eosinophils (Supplementary data 14). CellChat analysis showed the enrichment of IL4 signaling (Fig. 6f) and the predominance of *IL13-IL13RA1* ligand–receptor pairs (Fig. 6g) in blister samples. Although the *IL13* signals were also observed in Th2/Tc2 and Proliferating T, the eosinophils were found to be the main source of *IL13* signals, and DCs and macrophages were the prominent recipients in blister samples (Fig. 6g, Supplementary Fig. 12e, f). It has been noted that eosinophils have the potential to secret IL-13 and the eosinophils-derived IL-13 can promote Th2 polarization[50], which was also validated in our data the communication from eosinophils to Th2/Tc2 cells (Supplementary Fig. 12f, left column). Therefore, we hypothesized that the Th2 cell-mediated cascade responses facilitated pathogenic anti-BP180-NC16A autoantibodies, recruiting the infiltration of eosinophils and secretion of IL-13 to further amplify the type 2 inflammation.

In line with the lesional samples, although the expression level of CCL17 was relatively low in PBMC samples, the *CCL17-CCR4* ligand–receptor pair was also activated mainly outgoing from DCs to T cells. Intriguingly, B cells also represented as the source of CCL17 chemokine in PBMC samples (Supplementary Fig. 13a, b). CCL17 signaling also mediated the DC-T interaction in the blister samples (Supplementary Fig. 13c, d).

Collectively, we further validated the *IL13-IL13RA1* and *CCL17-CCR4* ligand–receptor pairs in PBMC and blister samples, establishing the precision of the observations in skin lesions.

## Discussion

Our study unveiled a positive feedback loop between immune cells and fibroblasts, leading to the gradual accumulation of Th2 cells and amplifying Th2 cell-mediated cascade responses. Additionally, we demonstrate fibroblast-produced PLA2G2A and CCL17 derived from myeloid cells play essential roles in promoting the secretion of pathogenic anti-BP180-NC16A autoantibodies (Fig. 7). These findings deepen our understanding of the pathogenesis of BP and other type 2 inflammation-driven conditions, while also revealing mechanisms within fibroblast and myeloid cells that may contribute to the development of these conditions.

Type 2 cytokines, specifically IL-4, IL-5, and IL-13, play essential roles in the immune-pathogenic mechanisms of type 2 immune response-driven diseases. Notably, different predominance of these type 2 cytokines is observed in type 2 inflammation-driven airway diseases and skin conditions. In asthma, these cytokines induce various key features such as IgE and IgG1 synthesis by B cells (IL-4), eosinophil recruitment (IL-5), and control of mucus production and bronchial hyper-responsiveness (IL-13)[51]. In type 2 inflammation-driven skin conditions, IL-13 appears to have a more prominent role[28]. This is supported by findings such as the predominance of *IL13* expression with very low *IL4* expression, and a strong correlation between IL-13 expression levels and disease severity in lesional skin from AD patients[29,30,52]. In BP, IL-13⁺ cells, but not IL-4⁺ cells, are correlated with itch severity[18]. Additionally, IL-13 gene polymorphism is significantly associated with the pathogenesis and recurrence of BP[10]. However, the mechanisms by which IL-13 exerts its effects and the detailed signaling networks involved remain poorly understood. In our study, the expression levels of IL-13 were more prominent than other type 2 cytokines in BP lesions, similar to observations in AD[29]. We identified the *IL13-IL13RA1* pair as the most significant ligand–receptor pair in IL4 signaling. Fibroblasts and DCs, as the primary recipients of IL-13 signals, responded to IL-13 derived from Th2 cells and established a positive feedback loop, amplifying Th2 cell-mediated immunity. This striking positive feedback loop of Th2-fibroblasts/DCs highlights the crucial role of fibroblasts in the pathogenesis of BP.

Fibroblasts are recognized as contributing to both skin homeostasis and pathophysiology in various ways. Different subpopulations of fibroblasts have been reported to recruit immune cells and influence neurite outgrowth and profibrotic responses by secreting chemokines (such as CCLs and CXCLs) and extracellular matrix proteins in skin conditions[33,36,38,53]. In our study, we identified two IL-13-responsive fibroblast subsets in skin lesions that recruit immune cells through the CXCL12-CXCR4 axis. This mechanism is reminiscent of the CXCL9/CXCL10–CXCR3 axis observed in vitiligo[36] and the CCL19-CCR7 axis in patients with AD[33]. These two fibroblast subsets highly expressed *PLA2G2A*, a gene encoding secretory calcium-dependent phospholipase A2. This enzyme primarily targets extracellular phospholipids and has implications in host antimicrobial defense, inflammatory response, and tissue regeneration[54]. PLA2G2A has been associated with poor survival in various cancers and has been implicated in the recruitment of immune cells[44,55–60]. However, its role in skin diseases has not been fully elucidated, except for the genetic deletion of PLA2G2A, which affects skin carcinogenesis and exacerbates psoriasis by shaping the

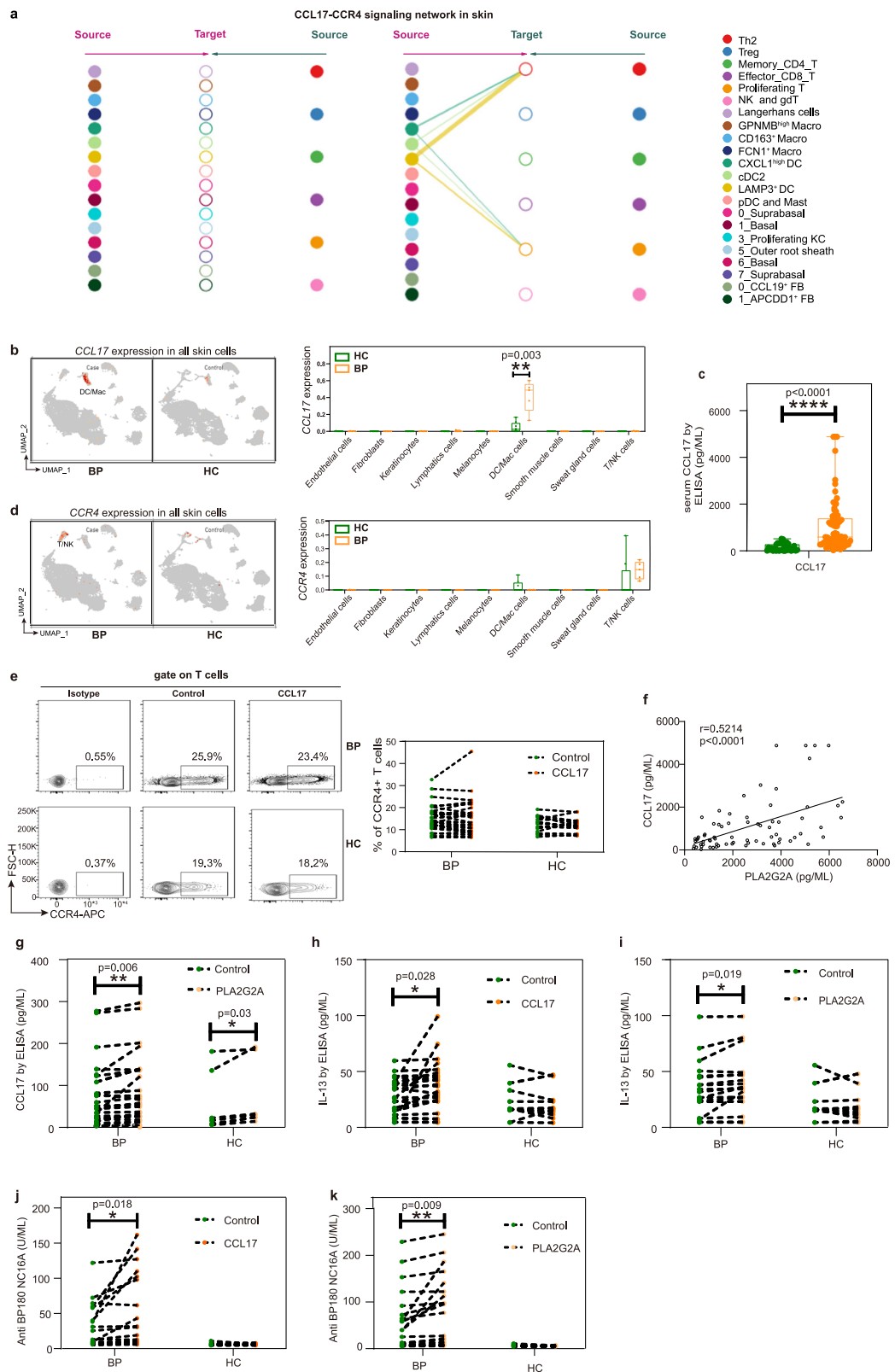

gut microbiota[61]. In our study, the CXCL12/CXCR4 axis was predominant, originating from the lesional fibroblast subpopulations overexpressing *PLA2G2A* to nearly all immune cells. Significant positive correlation between the expression of *PLA2G2A* and *CXCL12* was observed in fibroblasts, and PLA2G2A recombinant protein increased the expression of CXCR4 on T cells from PBMCs. This discovery reveals a mechanism underlying immune-stromal crosstalk that upregulation

of PLA2G2A mediates local aggregation of immune cells through CXCL12/CXCR4 axis from periphery blood into the lesional sites in type 2 inflammation-driven skin conditions, which is reminiscent of the fibroblasts-derived CXCL9/CXCL10–CXCR3 axis-mediated CD8[+] T cells infiltration in vitiligo[36]. Additionally, PLA2G2A increased the titer of anti-BP180-NC16A autoantibodies in PBMCs from BP patients, thereby amplifying the autoimmune response. Moreover, the CCL19-CCR7

**Fig. 5 | CCL17 is increased in IL-13-responsive myeloid cells in BP patients, and CCL17 and PLA2G2A promote the secretion of IL-13 and pathogenic anti-BP180-NC16A autoantibody. a** Hierarchical plot showing inferred intercellular communication network of *CCL17-CCR4* signaling. **b** The feature plot and the bar chart showing *CCL17* mRNA expression within total skin cells between BP (*n* = 5) and HC (*n* = 8). **c** Protein level of CCL17 in serum from BP (*n* = 73) and HC (*n* = 32) by ELISA. **d** The feature plot and the bar chart showing *CCR4* mRNA expression within total skin cells between BP (*n* = 5) and HC (*n* = 8). In the box plot **b**–**d**: Minima: Lower limit of the whisker. Maxima: Upper limit of the whisker. Center: Median line inside the box. The upper and lower box bounds represent the 25% and 75% percentile of data. **e** Flow plots of CD3$^+$ T cells from PBMCs showing the expression of CCR4 treated by CCL17 recombinant protein (left panel), and the frequency of CCR4 (right panel) in BP (*n* = 30) and HC (*n* = 16) groups. **f** The positive correlation between the level of PLA2G2A and CCL17 in serum from in BP patients (*n* = 73). *P*-value was calculated using two-sided Pearson correlation test. *r*-value was Pearson correlation coefficient. **g** The level of CCL17 in PBMC from BP patients (*n* = 26) and HC (*n* = 7) treated with PLA2G2A recombinant protein. **h** Effect of CCL17 treatment on the IL-13 secretion from BP patients (*n* = 26) and HC (*n* = 13). **i** Effect of PLA2G2A treatment on the IL-13 secretion from BP patients (*n* = 20) and HC (*n* = 13). **j**, **k** ELISA analysis of anti-BP180-NC16A antibody titers in supernatants of CCL17 (**j**) or PLA2G2A (**k**) stimulated PBMCs from BP patients (*n* = 26) and HC (*n* = 22). *P*-values in **b**–**d** were calculated using two-sided Mann–Whitney U-test. *P*-values in **e** and **g**–**k** were calculated using paired two-sided Student's *t*-test. \**P* < 0.05, \*\**P* < 0.01, \*\*\*\**P* < 0.0001, only *P*-values < 0.05 are shown. Each sample is represented as one dot.

axis, previously reported in AD patients[33], was found to be specifically active in BP patients, contributing to communication between fibroblasts (*CCL19*$^+$ FB) and dendritic cells (cDC2) in skin lesions, and from DC cells to nearly all immune cells in blister samples (Supplementary Fig. 14). Additionally, the extracellular matrix protein tenascin-C (TNC), known to be involved in neurite outgrowth in a mouse model of psoriasis[38], was significantly upregulated in BP lesional fibroblasts (Fig. 4a, b). Given that chronic itch is a hallmark of BP, TNC may play a role in regulating the cutaneous sensory nerves of BP patients. Further studies will be conducted to explore the role of TNC in BP. In summary, the identification of distinct fibroblast subsets in BP patients provides novel therapeutic possibilities for both type 2 inflammation-driven diseases and autoimmune bullous diseases.

Furthermore, in IL-13-responsive DC clusters, the levels of the type 2 chemokine CCL17 were significantly elevated in BP patients. CCL17, expressed by myeloid cells, plays a pivotal role in attracting Th2 cells into the skin, contributing to the development of various Th2-type inflammatory skin diseases as well as cutaneous lymphoma[62]. In BP, serum CCL17 levels have a positive correlation with BP Disease Area Index scores[45,46] and fluctuated in parallel with eosinophil counts[47]. Over the course of the disease, the level of CCL17 decreased in association with symptomatic improvement[63,64]. In our study, we discovered that fibroblast-derived PLA2G2A triggered the secretion of CCL17, as the stimulation of PLA2G2A recombinant protein elevated the levels of CCL17. We also observed that CCL17 stimulated IL-13 production and increased the titer of anti-BP180-NC16A autoantibodies in BP PBMCs. These findings elucidate the mechanism involved in the upregulation of CCL17 in type 2 inflammatory diseases and demonstrate the amplification of pathogenic autoantibodies by CCL17 in the mechanism of BP.

We acknowledge the limitations of our study. Firstly, in skin scRNA data, clusters for B cells and granulocytes were not observed, and only a small cluster of mast cells was identified. Subsequent work will further explore the role of innate immune cells in BP. Secondly, the activation of fibroblasts in skin diseases remains not entirely understood, despite previous work by Cai and colleagues, who observed inflammation-induced TNC overexpression in fibroblasts in a psoriasis mouse model[38].

In summary, our study provides several key findings. Firstly, we identified the *IL13-IL13RA1* as the primary ligand–receptor pair in IL4 signaling, promoting crosstalk between Th2 cells and fibroblasts/DCs, and amplifying Th2 cell-mediated responses. Additionally, upregulated PLA2G2A in lesional fibroblasts recruits immune cells via the CXCL12/CXCR4 axis, revealing a further mechanism in type 2 inflammation-driven skin conditions. Furthermore, fibroblast-derived PLA2G2A and myeloid-derived CCL17 facilitate the secretion of pathogenic anti-BP180-NC16A autoantibodies (Fig. 7), expanding our understanding of immune cells-fibroblasts crosstalk and offering insights into therapeutic avenues for type 2 inflammatory diseases.

## Methods

### Ethics statement

All procedures involving the use of human samples were conducted in compliance with the guidelines and regulations of the institutional ethics committee of Shandong Provincial Institute of Dermatology and Venereology. Moreover, written informed consent and permissions to collect blood, blister specimens, and skin lesions were obtained from all study participants or parents or guardians for those under 18.

### Study design

The objective of the study was to gain insight in the intricate interplay between immune and stromal cells of type 2 inflammatory diseases using Bullous pemphigoid (BP) patients, we employed single-cell RNA sequencing (scRNA-seq), cell-cell communication analysis and in vitro functional analysis on multiple sample types, including skin lesions, peripheral blood mononuclear cells (PBMC), and blister specimens obtained from patients with BP and healthy controls. Both blister samples and the surrounding erythema were collected for skin scRNA-seq. All participants were of Chinese descent, with a dermatologist-confirmed diagnosis of BP (Supplementary Table 1). Self-reported sex and ethnicity were recorded. In this study, we included a total of 135 patients with BP and 96 HC. The discovery cohort comprised 10 BP patients and 16 HC individuals. The first validation cohort included 31 BP patients and 22 HC, the second validation cohort consisted of 70 BP patients and 33 HC, and the third validation cohort included 53 BP patients and 29 HC. The age, sex and other clinical information of all samples are summarized in Supplementary Table 1.

Diagnosis of BP relied on typical clinical and histological presentations, alongside direct or indirect immunofluorescence examinations. It is important to note that all participants included in the study had not received any systemic or topical immunosuppressive treatments before the collection of samples. Control PBMC samples were obtained from healthy donors, and control skin biopsies were taken from patients with injury and without infectious or immune-related diseases.

### Single-cell preparation, RNA sequencing, and data analysis

Skin biopsy specimens were processed as follows: They were disassociated with Dispase II (Sigma) to separate the epidermis and dermis. The minced epidermis was further digested with 0.25% Trypsin-EDTA (Gibco) for 30 min and filtered using a 70 μm cell strainer (Falcon). The dermis was digested with 1 mg/mL Collagenase P (Sigma-Aldrich) and 100 μg/mL DNase I (Sigma-Aldrich) for 50 min and filtered through a 70 μm cell strainer (Falcon).

Barcode labeling of single cells and library construction were conducted using the 10× chromium system (10× genomics). The constructed library was subsequently sequenced on an Illumina NovaSeq 6000 System. For each sample dataset, the raw sequencing data were aligned and quantified using the CellRanger pipeline (version 6.1.2, 10× Genomics) against the GRCh38-2020-A human reference genome.

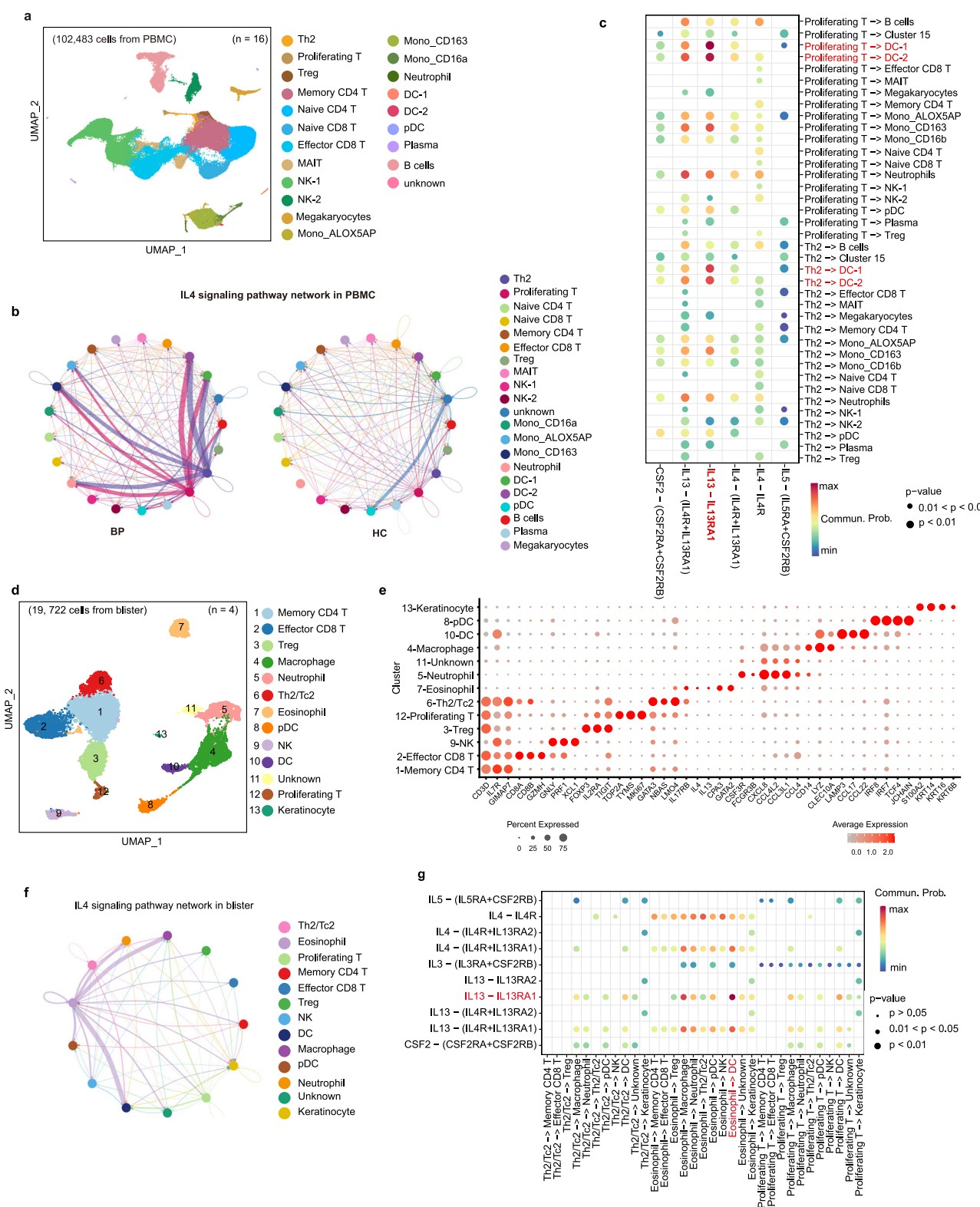

**Fig. 6 | *IL13-IL13RA1* is the most significant ligand–receptor pair among IL4 signaling pathways in PBMC and blister scRNA-seq datasets. a** UMAP visualization of the PBMCs scRNA-seq profile. **b** Circle plot of the inferred IL4 signaling pathway in PBMC samples from the BP and HC groups. **c** Comparison of the multiple ligand–receptor pairs among IL4 signaling pathway in PBMCs. **d** UMAP visualization of the cells from blister scRNA-seq profile. **e** Expressions of major discriminative marker genes for cell types identification of blister scRNA-seq dataset. **f** Circle plot of the inferred IL4 signaling pathway in blister samples from the BP group. **g** Comparison of the multiple ligand–receptor pairs among IL4 signaling pathway in blister cells. *P*-values in **c** and **g** were computed from one-sided permutation test (CellChat 1.5.0).

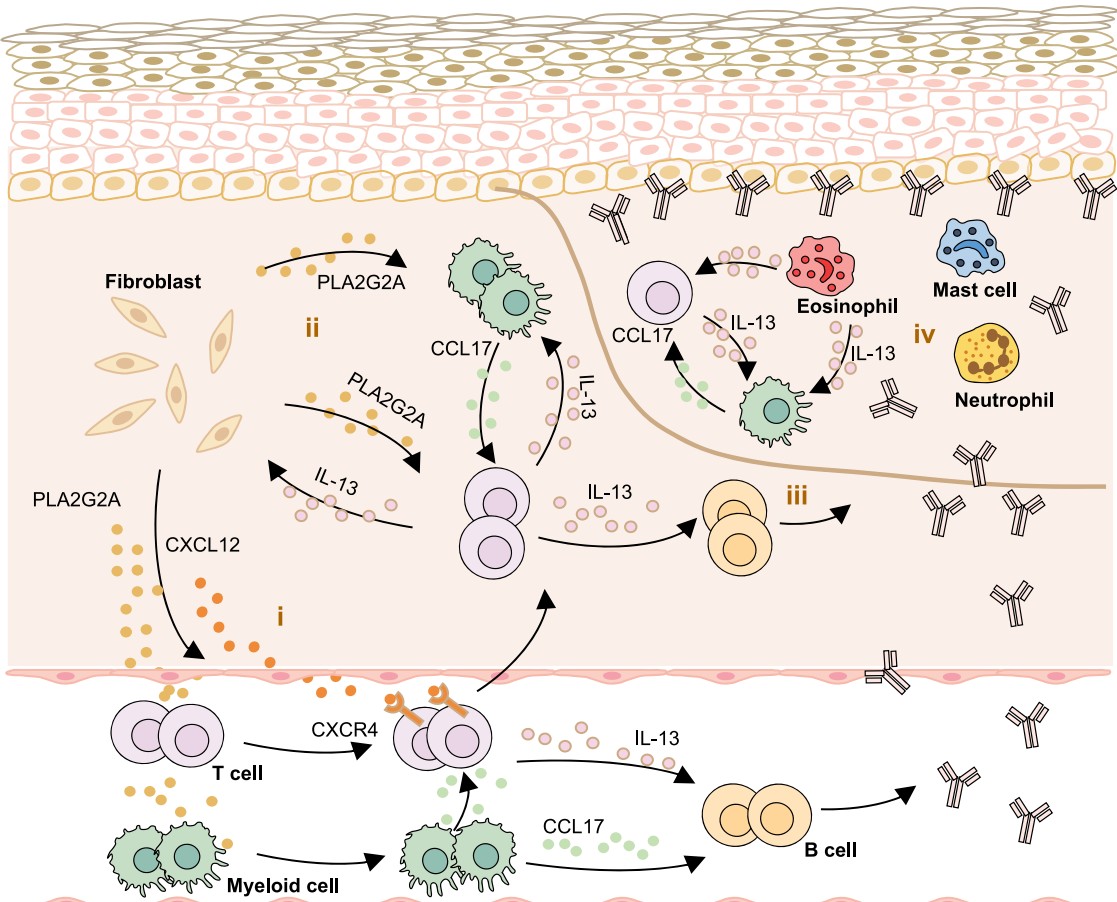

**Fig. 7 | Schematics of the immune-stromal crosstalk in BP patients.** It illustrates a positive feedback loop where fibroblasts respond to Th2 cells through the *IL13RA1-IL13* pair, leading to increased secretion of PLA2G2A and CXCL12. This, in turn, amplifies the Th2-mediated response. (i) The lesional fibroblasts respond to IL-13 and induce the overexpression of PLA2G2A, which promotes the expression of CXCR4 on the surface of immune cells to recruit the immune cells from peripheral blood into skin lesions. (ii) Fibroblasts-derived PLA2G2A and myeloid cells-derived CCL17 elevate the secretion of IL-13, and fibroblast and myeloid cells further respond to IL-13, forming a positive feedback loop between immune cells and fibroblasts. (iii) IL-13 activates B cells to secrete autoantibodies. (iv) In blister, the secretion of autoantibodies recruits T cells, myeloid cells, mast cells, neutrophils and eosinophils. T cell-derived IL-13 activates eosinophils to secrete various cyto-kines (including IL-13), which further promotes Th2 polarization and mediates the crosstalk between immune cells.

The gene expression matrix was then subjected to processing and analysis using the Seurat (version 4.0.0) R package[65] (version 4.1.0). Cells that fell below specific quality thresholds were removed: those with a total unique molecular identifier number fewer than 500, detected with fewer than 200 genes, or detected with more than 10,000 genes. Cells with more than 10% mitochondrial reads or more than 10% red cell reads were also excluded. Additionally, cells with more than 10% mitochondrial reads predicted by mitoRiboUmi func-tion of rCASC package (https://kendomaniac.github.io/rCASC/reference/mitoRiboUmi.html)[66] and cells predicted to be potential doublets using the Python Scrublet package (version v0.2.3)[67] were excluded.

Following filtering, the gene–barcode matrix was initially nor-malized using Seurat's "NormalizeData" function with default para-meters. The top 2000 highly variable genes were used to remove batch effects and perform further dimensionality reduction via the "Run-Harmony" and "RunUMAP" functions, respectively. The Uniform Manifold Approximation and Projection (UMAP) was employed for visualization. Clusters were calculated using the "FindNeighbors" and "FindClusters" functions with specific resolution parameters (0.2 for the clustering of all skin cells, 1.4 for the clustering of myeloid cells, 0.8 for the clustering of fibroblasts, 0.2 for the clustering of keratinocytes, 0.1 for the clustering of all PBMC cells and 0.4 for all blister cells).

These calculations were based on the top 30 principal components from Harmony. Cluster marker genes were identified using the "Fin-dAllMarkers" function with a filter condition of log2 (Foldchange) (log2FC) > 1 and adjusted $P$-values < 0.05. In the sub-clustering of immune cells and fibroblasts, potential doublets were identified and subsequently removed from further analysis. Differential gene expression analysis between the cases and controls was performed using the "FindMarkers" function with a filter condition of |log2FC| > 1 and adjusted $P$-values < 0.05. The annotation of clusters was validated by overlaying the cluster marker genes with canonical signature genes that define cell types.

**Cell-cell interaction analysis**

To explore cell-cell interactions between different cell types, we employed the CellChat R package (version 1.5.0), which is based on known interactions between signaling ligands, receptors, and their cofactors as described by Jin. et al[68]. Initially, we used CellChat to assess the significance of signaling pathways based on the differences in overall information flow within the inferred networks between BP and control groups. Subsequently, the netVisual_circle function was utilized to visualize the strength or weakness of cell-cell communica-tion networks from the target cell cluster to various cell clusters within the significant signaling pathways. Additionally, we employed the

netVisual_bubble function and the netVisual_hierarchy function to create bubble plots and hierarchical plots that represent significant ligand–receptor interactions between the target cell cluster and other clusters.

## Immunohistochemical staining and immunofluorescence

For immunohistochemistry, we first blocked endogenous peroxidase in paraffin-embedded tissue sections using 3% $H_2O_2$. Non-specific binding was then blocked with a 5% bovine serum albumin blocking buffer (Solarbio) after performing citrate antigen retrieval. Subsequently, the sections were incubated with anti-CD3 antibody, anti-CD68 antibody, anti-CD11c antibody, anti-GATA3 antibody, anti-IL-13 antibody, anti-PDGFRα antibody, anti-PLA2G2A antibody, anti-CXCL12 antibody, anti-CXCR4 antibody, and anti-CD207 antibody (Supplementary Table 2) at 4 °C overnight. Following this, HRP-labeled secondary antibodies were applied for 60 min at room temperature. DAB visualization (ZSGB-BIO, ZLI-9018) was used to detect the biotinylated antibodies.

For immunofluorescence staining, the slides were incubated with primary antibodies overnight at 4 °C. This was followed by incubation with secondary antibodies, and color development was carried out using a Four-color multiple fluorescent immunohistochemical staining kit (Absin, ab50012). The primary antibodies used for multicolor immunohistochemistry included anti-human CD3, anti-human GATA3, anti-human CD68, and anti-human PLA2G2A to observe protein expression and localization. Images were acquired using the EVOSTM FL Auto 2 Imaging System (Thermo Fisher) or the LSM 980 confocal microscope (ZEISS).

## ELISA

The serum levels of IL-13 (EK0424), IL-4 (EK0404), IL-5 (EK0407), PLA2G2A (EK1944), and CCL17 (EK0684) in both patients and controls were quantified using ELISA Development Kits from BosterBio, USA, following the manufacturer's instructions. To determine antibody levels in cell-culture supernatants, we used Anti-BP180 and Anti-BP230 antibody ELISA kits from Euroimmune, Germany, in accordance with the manufacturer's instructions.

## Chemotaxis assay

A total of $5 \times 10^5$ THP1 or Jurkat cells from ATCC were loaded into the top chamber of 24-well tissue culture inserts (Costar). PLA2G2A (0.5 µg/ml, 11187-H08H, Sino Biological) was separately applied to the top chamber in RPMI-1640 and the bottom chamber in RPMI-1640 containing 20% serum. After incubation at 37 °C in 5% $CO_2$ for 4 h, the cells that had migrated into the bottom chamber were collected and counted.

## In vitro stimulation

PBMCs or sorted T cells or DCs from BP patients and HC were plated at a concentration of 1*10^6 cells per well in 1000 µl RPMI supplemented with 10% FBS. These cells were then stimulated with or without PLA2G2A (0.5 µg/ml, Sino Biological) and CCL17 (100 ng/ml, R&D). After 7 days, the cells were washed and resuspended in staining buffer (PBS with 2% FBS) and subjected to flow cytometry analysis to assess the expression of CXCR4 and CCR4 on T cells, and CD138 on B cells.

## Flow cytometry

After 7 days, the stimulated PBMCs were incubated with surface antibodies (CD3, CD19, CXCR4, CCR4, and CD138) in a staining buffer for 30 min at 4 °C. Data were collected on the FACSAria Fusion flow cytometer (BD Biosciences) and analyzed using FlowJo software. The following conjugated monoclonal antibodies (mAbs) were used: CD3 (OKT3), CD19 (SJ25C1), CD138 (DL-101), CXCR4 (12G5), CCR4 (L291H4) (Supplementary Table 2). Isotype controls were used for gating.

All mAbs were purchased from eBioscience, Biolegend, or BD Bioscience. The gating strategy was provided in Supplementary Fig. 15.

## Reporting summary

Further information on research design is available in the Nature Portfolio Reporting Summary linked to this article.

## Data availability

The raw scRNA-seq data of 33 samples in fastq format generated in this study have been deposited in the Genome Sequence Archive (GSA) database under accession code HRA005922, HRA003993, HRA000145, HRA000471, HRA000847, HRA005913 and HRA003995 [https://ngdc.cncb.ac.cn/gsa-human]. Among these, twenty-five samples are new additions, including four blisters and eight PBMCs from BP patients (HRA005922), five lesions from BP patients (HRA003993), three samples of normal skin from HC (HRA003995), and five PBMCs from HC (HRA005913). Furthermore, eight samples have been previously reported: three samples of normal skin from HC (HRA000145), two more samples of normal skin from HC (HRA000471), and three PBMCs from HC (HRA000847). The scRNA-seq data of these 33 samples in 10X Genomics format have been also deposited in Zenodo database under accession code 10924853. These data are freely available without any restrictions or time limits. All other data are available in the article and its Supplementary files or from the corresponding author upon request. Source data are provided with this paper.

## Code availability

Representative code is available on GitHub (https://github.com/zzwang1030/scRNA_BP, https://doi.org/10.5281/zenodo.11567155). This code is openly available with no restriction or time limit. Any queries or further requests can be addressed to the corresponding authors.

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

## Acknowledgements

This work was supported by grants from the Central guidance for local scientific and technological development projects of Shandong Province (YDZX2023058, HL), the Key research and development program of Shandong Province (2021LCZX07, FZ), National Natural Science Foundation of China (82304039, TL) and the Shandong Province Taishan Scholar Project (tsqn201909141, YS). We would like to thank Dr. Xiaojiang Xu and Dr. Jianan Li from Wanhui Bioinformatics Team and Dr. Ziwei Wang from Genenergy Bio-Technology, for the help in data analysis. We are grateful to the patients and healthy controls for donating skin tissue or blood or blister.

## Author contributions

HL and FZ designed this study. TL, XX, QZ, GY and MW collected the clinical samples. TL and Zhenzhen Wang analyzed single-cell-sequencing datasets. TL, Zhe Wang, YZ, ZM, CW, LS and PS performed biological Experiments. TL and Zhenzhen Wang wrote the manuscript. HL, YS, FX and FZ revised the manuscript.

## Competing interests

The authors declare no competing interest.
