## [Peer Review File · Nature Communications]

Single-cell transcriptomics analysis of bullous pemphigoid unveils immune-stromal crosstalk in type 2 inflammatory diseaseREVIEWER COMMENTS

Reviewer #1 Bullous Pemphigoid (Remarks to the Author):

Tingting Liu et al. identified a potential local inflammatory pathway in bullous pemphigoid by analyzing single cell RNA sequencing data from patient skin and PMMC samples. The authors partially validated their finding with in vitro analysis.

The authors demonstrated increased immune cell populations in BP lesions particularly T/NK and DC/Macs. Focusing on these cell types, the authors revealed distinct sub clusters and their proportional change in BP versus healthy skin, which lead to the authors suggestion of a type 2 immune environment, which has been described in literature previously. Moreover, they compared non-immune populations focusing on keratinocytes and fibroblasts. Using CellChat analysis on their identified BP related clusters (immune / non-immune), the authors determined the probability of these cell populations to interact with each other.

They identified the IL13-IL13RA1 axis as one of the most significant signaling pathway within their analysis, which correlated with two of the identified fibroblast, and four of the myeloid clusters. Further analysis revealed upregulation of PLA2G2A in both of the fibroblast's clusters, which in turn was further explored regarding expression by ELISA in BP serum samples. By further cell cluster analysis of BP vs. healthy skin and PBMCs they propose a possible mechanism by activation of CXCL12-CXCR4 axis originating from fibroblasts and CCL17-CCR4 derived from fibroblasts and myeloid cells respectively to induce activation of autoantibodies in PBMCs. However, as most of the data are only partially validated in vitro, the final mechanism is still elusive.

Overall, the data are analyzed straight forward and are well presented. They significantly contribute to our knowledge in skin autoimmunity and especially the pathogenesis of BP. However, I have some concerns in interpretation and validation of sc-data:

1. The authors do not provide information on the control skin and blood donors with regard to age and sex. Have the authors matched these to the patients? According to the methods section, all patients were diagnosed with BP based on clinical and histological presentation as well as autoantibody titers against BP180-NC16A. Autoantibodies against BP230 were accessed, however the data was not stratified regarding singular (BP180) or dual (BP180, BP230) reactivity. Strikingly this singular / dual reactivity of patient sera was not taken into account for the following validation cohorts, where the balance of patients is not matched with the discovery cohort. The reviewer understands, that the collection of samples is challenging, however with regard to the journal guidelines on sex distribution, have the authors tried to stratify their data in this regard? From supplement table 1 it is clear that an almost 4:1 ratio of M:F samples was used in this study.
2. A majority of patients included in the validation cohort 3 do not present with anti-BP230 antibody titers, whereas a majority of the discovery cohort does show reactivity to BP230. How might this affect the authors conclusions?
3. Table S1 lacks units– I am assuming – anti-BP180 / 230 titers?
4. Figure 1e. The data must be interpreted with caution, i.e., older publications show an

increase of Langerhans cells in BP (*Acta Derm Venereol.* 1987;67(6):529-32). Maybe more IF stainings (CD206 and others) would help to confirm the sc-data.

5. Figure 1f. Staining of CD3/GATA shows an overall increase of CD3 T cells that are not clearly characterized and/or discussed. To confirm that specifically Th2 cells increase, quantification and staining of further cell markers would be interesting (f.e. T-bet, Foxp3 and/or CD25). For a better understanding of the skin architecture it would be nice to include a line or arrowheads at the DEJ (same to Figure 4f).

6. Figure 2a. The Flow is very complex and the abbreviations should be given in the supplement. It is unclear to me, why the focus was given to IL4 exclusively as this pathway was already described before in BP patients whereas others are not.

7. Figure 4f. In the skin staining PLA2G2A seems to be exclusively cytosolic, which is not in line with the later interpretations. In addition, it would be nice to see a double staining with the respective fibroblast markers to confirm the scRNA data. Adding the DEJ by lines or arrows would be helpful to understand the skin orientation. "Immunofluorescence analysis showed that PLA2G2A+ fibroblasts had a similar spatial distribution to CD68+ macrophages and CD3+ T cells" – For me it looks rather that macrophages concentrate near the DEJ (to the left?) whereas T cells are located in lower layers?

8. Figure S4. For me the further investigation of the CXCL12 / CXCR4 pathway is somehow unclear. Looking to the analysis, i.e., MIF seems to be more interesting. What was the reason to focus on CXCL12 / CXCR4? Again, IF stainings would be interesting for confirmation of the proposed hypothesis.

9. Figures 4i and j. The data seem a bit overinterpreted to me as there is no obvious difference between BP and controls. In figure 4i, the difference is probably given by the fact that PLA2G2A is expressed in BP only? Again, double staining would be interesting to confirm the scRNA data.

10. Figure 5. Similarly to figure 4, the most differences in ELISAs are given by 3 patients that show an effect whereas all other patients do not? Did you check for BP230 level, age and sex here? Maybe this is just a sub-cohort? In addition, in figure 5j/k no effect for HC is expected as the autoreactive cells are missing.

11. Figure 5g. "These results indicated that the fibroblast-derived PLA2G2A may drive the secretion of CCL17 from myeloid cells, especially the DC clusters, in BP patients." This seems speculative. To confirm, you should include in vitro experiments. Overall, most (not all) interpretations focus on RNA data or ELISA from serum. To confirm the proposed mechanism, the respective cell types (f.e. fibroblasts, myeloid cells) could be stimulated in vitro and the target proteins could be detected on protein levels by WB, IF or FACS. If not possible, this limitation of the study has to be clearly discussed.

12. Figure 5j/k. The effect of CCL17 and PLA2G2A on autoantibody production is very interesting. However, no B cells are found in the clusters in the skin (figure 1B). Therefore, another effect seems to be important in the skin. Can you comment on this? (Just a comment: Furthermore, no clusters for granulocytes and mast cells are visible in figure 1B - which I know is technically very challenging. However, all interpretation made by lesional scRNA data here miss the effect of these important innate immune cells).

13. With figure 7, the authors provide a scheme how they envision the interaction circle in BP. To the reviewer this scheme is somewhat misleading, as from the current depiction, the release of PLA2G2A and CXCL12 seems to trigger the initial response. Furthermore, (i) focus only on IL-13 (not IL-4) is somehow overinterpreted (ii) what B cells / plasma cells are known to be in the skin in BP patients? (iii) is PLA2G2A really secreted by skin fibroblasts (see figure 3f)? (iv) according to figure 3f the source of IL13 is Th2 and pDC/Mast and proliferating T cells?

Minor comments: The authors tend to switch between stromal cells and fibroblasts, which to my understanding they use synonymously. The manuscript is overall understandable when reading carefully. However, the text would benefit from professional revision regarding formulation and grammar, easing up longer paragraphs and choice of appropriate, scientific wording. Some terms like IL4/IL-4 are not used consistently. Also the writings for RNA and proteins are not consistent.

Reviewer #2 skin inflammation (Remarks to the Author):

This study performed the scRNA-seq using lesional skin samples, PBMCs, and blister fluids of BP and demonstrates novel findings regarding the crosstalk between fibroblasts and immune cells, that may lead to a better understanding of the pathogenesis of BP and the development of new therapies. I have the following comments which may be helpful for the improvement of the work.

- 1) What kind of lesions were obtained for scRNA-seq analysis, blister or erythema? It should affect the characteristics of fibroblasts and immune cells. Please clarify it.
- 2) Langerhans cells often play an important role in the immune response in the skin. Fig1e shows a lower percentage of Langerhans cells in BPs compared to HCs. On the other hand, Fig3e shows that IL13-IL13RA1 is very high in Langerhans cells. It is better to mention the significance of Langerhans cells in the discussion.
- 3) Please explain in detail how to read the hierarchical plot showing inferred intercellular communication network (Fig 3f, 4h, and 5a) in the figure legends or method section, e.g., what do the two opposing arrows at the top of the figure mean?
- 4) Fig 4e shows the function of PLA2G2A on monocytes using THP1, but it is better to also look at the function of PLA2G2A on T cells.
- 5) The results in Fig 4d are interesting. Please show the correlation between serum PLA2G2A level and BPDAl.
- 6) Regarding Fig 5g, h, i, it is unclear which cells PLA2G2A or CCL17 worked on since you are using PBMC. You should do the experiment with PBMCs from which DCs and/or T cells were removed to confirm that they are really DCs or T cells.
- 7) Regarding Fig 5j and k, it is difficult to understand why anti-BP230 antibodies are not elevated by PLA2G2A and CCL17. Only samples from patients who were positive for anti-BP230 antibodies should be re-examined.

Reviewer #3 scRNA and systems (Remarks to the Author):

I conducted a thorough examination primarily centered on data and methods, with a specific focus on assessing quality of the analyses and the reproducibility of the procedures outlined in the current manuscript.

Major points:

1) While the procedure outlined for single-cell data processing is reasonably clear, to guarantee reproducibility, it is essential for the authors to share the code used for all the bioinformatics steps mentioned in the Method section, including single-cell preparation, RNA sequencing, and data analysis, on a GitHub repository.

Moreover, the code used to generate Figure 1b-d, Figure 2a-f, Figure 3a-f, Figure 4a-e,g-j, Figure 5a-k, Figure 6a-g must be provided. Same apply to supplementary figures. The code must be linked to the data used in the analysis, which must be provided as part of the github or as zenodo/figshare repository.

2) In Method section Single-cell preparation, RNA sequencing, and data analysis the authors wrote: "Cluster marker genes were identified using the "FindAllMarkers" function with a filter condition of $\log_2(\text{Foldchange}) (\log_2\text{FC}) > 0.25$ and adjusted P values < 0.05 . In the subclustering of immune cells and fibroblasts, potential doublets were identified and subsequently removed from further analysis. Differential gene expression analysis between the cases and controls was performed using the "FindMarkers" function with a filter condition of $|\log_2\text{FC}| > 0.585$ and adjusted P values < 0.05 ."

The authors should provide justification for selecting $\log_2\text{FC}$ values of only 0.25 and 0.585. These values are exceptionally small, and the authors must illustrate that they exceed the background levels in those experiments.

3) I remain unconvinced about the methodology employed in the Cell-Cell Interaction Analysis section. While CellChat serves as a tool for assessing cell-to-cell communication, its application to skin lesions with distinct multi-layer structure does not ensure that the identified cell-to-cell communication is occurring between cells that are truly in contact.

4) Cell type annotation must be better described. Specifically is not clear how the different cell types were defined.

5) The analysis of the integrated data is not sufficient, the analysis of the cell composition of each sample must be provided. Annotation must be done on each individual sample and eventually overlaid to the integrated data.

5) Metadata containing the cell type annotation associated to the corresponding single cell barcode must be provided for all the cells used in this manuscript.

6) Sparse count tables in 10XGenomics format (barcodes.tsv.gz features.tsv.gz matrix.mtx.gz) must also be provided in a figshare/zenodo repository.

7) The overall quality of each single cell RNAseq must be discussed in supplementary data, e.g. using the mitoRiboUmi function available as part of the rCASC package (<https://kendomaniac.github.io/rCASC/reference/mitoRiboUmi.html>)

REVIEWER COMMENTS

Reviewer #1 Bullous Pemphigoid (Remarks to the Author):

Tingting Liu et al. identified a potential local inflammatory pathway in bullous pemphigoid by analyzing single cell RNA sequencing data from patient skin and PMMC samples. The authors partially validated their finding with in vitro analysis. The authors demonstrated increased immune cell populations in BP lesions particularly T/NK and DC/Macs. Focusing on these cell types, the authors revealed distinct sub clusters and their proportional change in BP versus healthy skin, which lead to the authors suggestion of a type 2 immune environment, which has been described in literature previously. Moreover, they compared non-immune populations focusing on keratinocytes and fibroblasts. Using CellChat analysis on their identified BP related clusters (immune / non-immune), the authors determined the probability of these cell populations to interact with each other.

They identified the IL13-IL13RA1 axis as one of the most significant signaling pathway within their analysis, which correlated with two of the identified fibroblast, and four of the myeloid clusters. Further analysis revealed upregulation of PLA2G2A in both of the fibroblast's clusters, which in turn was further explored regarding expression by ELISA in BP serum samples. By further cell cluster analysis of BP vs. healthy skin and PBMCs they propose a possible mechanism by activation of CXCL12-CXCR4 axis originating from fibroblasts and CCL17-CCR4 derived from fibroblasts and myeloid cells respectively to induce activation of autoantibodies in PBMCs. However, as most of the data are only partially validated in vitro, the final mechanism is still elusive.

Overall, the data are analyzed straight forward and are well presented. They significantly contribute to our knowledge in skin autoimmunity and especially the pathogenesis of BP. However, I have some concerns in interpretation and validation of sc-data:

1. The authors do not provide information on the control skin and blood donors with regard to age and sex. Have the authors matched these to the patients?

According to the methods section, all patients were diagnosed with BP based on clinical and histological presentation as well as autoantibody titers against BP180-NC16A. Autoantibodies against BP230 were accessed, however the data was not stratified regarding singular (BP180) or dual (BP180, BP230) reactivity. Strikingly this singular / dual reactivity of patient sera was not taken into account for the following validation cohorts, where the balance of patients is not matched with the discovery cohort.

The reviewer understands, that the collection of samples is challenging, however with regard to the journal guidelines on sex distribution, have the authors tried to stratify

their data in this regard? From supplement table 1 it is clear that an almost 4:1 ratio of M:F samples was used in this study.

Reply:

(1) We appreciate your suggestions. At this juncture, enrolling new samples for the matching process in the discovery cohort poses significant challenges. Therefore, we have focused on augmenting the validation stage by including 78 new BP patients and 40 new controls. This step ensures the matching between patients and controls. With the increased sample size, we've achieved balanced gender distribution not only between BP and control groups but also within each validation cohort, where the M:F ratio among BP patients is now 1:1, as illustrated in **Reviewer Table 1**.

Reviewer Table 1. The gender information in each validation cohorts.

Characteristics	Validation cohort 1 (IHC, IF)		Validation cohort 2 (ELISA)		Validation cohort 3 (Flow)	
	BP	control	BP	control	BP	control
male	16	11	35	18	29	15
female	15	11	35	15	24	14

Acknowledging the slight age disparity between the control and case groups, despite our efforts to include older controls as detailed in **Reviewer Table 2**, we conducted a stratified analysis within the BP group to evaluate the potential impact of age on the main findings. We categorized the BP group into two age brackets: younger (Group1, <65 years old) and older (Group2, >=65 years old). Upon comparison, we found no significant differences in the main results between these two age groups, as depicted in **Reviewer Figure 1**.

Reviewer Table 2. The age information in each validation cohorts.

Characteristics	Validation cohort 1 (IHC, IF)		Validation cohort 2 (ELISA)		Validation cohort 3 (Flow)	
	BP	control	BP	control	BP	control
Age, y (mean±SD)	68.6±15.7	56.5±13.3	68.9±12.5	52.1±11.6	65.8±15.4	55.5±12.0

Reviewer Figure 1. (a, b, c) The levels of IL-13, PLA2G2A and CCL17 in serum. (d) The CCL17 secretion after PLA2G2A stimulation. The IL-13 secretion after CCL17 (e) or PLA2G2A (f) stimulation. The Anti-NC16A secretion after CCL17 (g) or PLA2G2A (h) stimulation.

In conclusion, we acknowledge the limitations stemming from the age and gender mismatches observed in the discovery stage. However, upon comparing our results with newly acquired samples to the previous findings in the validation cohort, it becomes evident that the main conclusions remain unaffected by gender or age. Therefore, we maintain the solidity of our conclusions.

(2) Furthermore, in order to maintain balance among patients in both the discovery and three validation cohorts concerning antibody reactivity, we included 78 new BP samples categorized into dual reactivity (22 patients), singular BP180 reactivity (31 patients), and singular BP230 reactivity (25 patients) in the validation cohorts. As anticipated, our primary findings remained unaffected by antibody reactivity.

Comprehensive information regarding the samples is provided in **Supplementary Table 1**.

2. A majority of patients included in the validation cohort 3 do not present with anti-BP230 antibody titers, whereas a majority of the discovery cohort does show reactivity to BP230. How might this affect the authors conclusions?

Reply: Thanks for your suggestion. To address the imbalance, we included seven new patients with dual reactivity (BP180, BP230) and 11 new patients with singular reactivity (BP230) in validation cohort 3.

The results remained consistent. We observed increased expression of CXCR4 on CD3+ T cells stimulated by PLA2G2A (**Reviewer Figure 2a**), comparable expression

of CCR4 on T cells stimulated by CCL17 (**Reviewer Figure 2b**), a significant increase in the proportion of CD138+ plasma cells of total B cells triggered by PLA2G2A or CCL17 (**Reviewer Figure 2c, d**), a pronounced increase in the secretion of IL-13 from BP cases after stimulation of PLA2G2A or CCL17 (**Reviewer Figure 2e, f**), and markedly increased production of anti-BP180-NC16A antibody in PLA2G2A- or CCL17-stimulated patient-derived PBMCs (**Reviewer Figure 2g, h**). Additionally, PLA2G2A drove the secretion of CCL17 (**Reviewer Figure 2i**).

In the revised manuscript, we have updated the corresponding figures accordingly.

Reviewer Figure 2. The results in validation cohort 3 after increasing seven new patients with dual (BP180, BP230) reactivity. (a, b) expression of CXCR4 and CCR4 on T cells by flow cytometry, (c, d) expression of CD138 on B cells by flow cytometry, (e, f) IL-13 secretion, (g, h) anti-BP180-NC16A antibody production, (i) CCL17 secretion.

3. Table S1 lacks units – I am assuming – anti-BP180 / 230 titers?

Reply: We have added the unit showing the anti-BP180 / 230 titers in Supplementary Table 1 accordingly.

4. Figure 1e. The data must be interpreted with caution, i.e., older publications show an increase of Langerhans cells in BP (*Acta Derm Venereol.* 1987;67(6):529-32). Maybe more IF stainings (CD206 and others) would help to confirm the sc-data.

Reply: We conducted new staining as per your suggestion (**Reviewer Figure 3, new Supplementary Fig. 2g**). The results reaffirmed an increase in the number of Langerhans cells observed via immunohistochemistry staining in BP patients, which aligns with findings from previous publications.

Reviewer Figure 3. The Langerhans cells in control and BP skin samples. CD207 were used to identify Langerhans cells based on published papers (*J Allergy Clin Immunol.* 2016; 138(5): 1436–1439. e11; *Immunity.* 2021;54(10):2305-2320.e11).

Additionally, in our scRNA-seq data, we observed a trend of increased absolute numbers of Langerhans cells in BP (**Reviewer Figure 4**). However, the proportion of Langerhans cells among myeloid cells in BP decreased compared to normal controls, primarily due to the marked elevation of DCs.

We have added a description in the result part: “Due to the marked elevation of DCs, the proportion of Langerhans cells among myeloid cells in BP decreased compared to normal controls. Actually, an increase in the number of Langerhans cells was observed via immunohistochemistry staining in BP patients, which aligns with findings from previous publications, indicating the significance of Langerhans cells in BP pathogenesis. ” (Line 132-136) and cited this paper (*Acta Derm Venereol.* 1987;67(6):529-32) in our revised manuscript.

Reviewer Figure 4. The absolute number of Langerhans cells in the skin RNA-seq data.

Reference:

1. Perelygina L, Plotkin S, Russo P, Hautala T, Bonilla F, Ochs HD, Joshi A, Routes J, Patel K, Wehr C, Icenogle J, Sullivan KE. Rubella persistence in epidermal keratinocytes and granuloma M2 macrophages in patients with primary immunodeficiencies. *J Allergy Clin Immunol.* 2016 Nov;138(5):1436-1439.e11. doi: 10.1016/j.jaci.2016.06.030. Epub 2016 Sep 6. PMID: 27613149; PMCID: PMC5392721.
2. Liu X, Zhu R, Luo Y, Wang S, Zhao Y, Qiu Z, Zhang Y, Liu X, Yao X, Li X, Li W. Distinct human Langerhans cell subsets orchestrate reciprocal functions and require different developmental regulation. *Immunity.* 2021 Oct 12;54(10):2305-2320.e11. doi: 10.1016/j.immuni.2021.08.012. Epub 2021 Sep 10. PMID: 34508661.
3. Emtestam L, Hovmark A, Lindberg M, Asbrink E. Human epidermal Langerhans' cells in bullous pemphigoid. *Acta Derm Venereol.* 1987;67(6):529-32. PMID: 2451382.

5. Figure 1f. Staining of CD3/GATA shows an overall increase of CD3 T cells that are not clearly characterized and/or discussed. To confirm that specifically Th2 cells increase, quantification and staining of further cell markers would be interesting (f.e. T-bet, Foxp3 and/or CD25). For a better understanding of the skin architecture it would be nice to include a line or arrowheads at the DEJ (same to Figure 4f).

Reply: Thanks for your suggestions.

(1) In **new Supplementary Fig. 1g**, which represents cluster 13 of the skin scRNA-seq data, we have demonstrated a general increase in CD3 T cells among BP patients compared to controls. Furthermore, we validated this finding through immunocytochemistry, as depicted in **new Supplementary Fig. 1h**. This result was emphasized in the results section as: “Cell composition analysis revealed an increase in the proportion of T/NK and DC/Mac cells in lesional skin relative to healthy control (Supplementary Fig. 1g). The increase of CD3+ T cells and CD68+ macrophages within the lesion of BP patients were confirmed by immunochemistry staining (Supplementary Fig. 1h).” (Line 99 - 102).

(2) The elevated presence of Th2 cells was further corroborated, as per your suggestion, through immunofluorescence staining of CD3/IL-13 (**Reviewer Figure 5, new Supplementary Fig. 2f**). Given the prominent expression of IL-13, the predominant type 2 cytokine observed in the skin scRNA-seq data, it was selected as the defining marker for Th2 cells. It's noteworthy that in defining classical lineage-specific transcription factors for Th1, Th2, Th17, and Treg cells—T-bet, GATA3, ROR γ t, and FOXP3 respectively (as outlined in *Nat Rev Immunol.* 2023; 23(12): 842- 856)—we utilized GATA3 in our previous manuscript to define Th2 cells.

Reviewer Figure 5. The IF staining of CD3/IL-13 in control and BP skin samples.

(3) A line at the DEJ have been included to better understand the skin architecture in **Figure 1f** and **Figure 4f**.

Reference

1. Trujillo-Ochoa JL, Kazemian M, Afzali B. The role of transcription factors in shaping regulatory T cell identity. *Nat Rev Immunol.* 2023 Dec;23(12):842-856. doi: 10.1038/s41577-023-00893-7. Epub 2023 Jun 19. PMID: 37336954; PMCID: PMC10893967.

6. Figure 2a. The Flow is very complex and the abbreviations should be given in the supplement. It is unclear to me, why the focus was given to IL4 exclusively as this pathway was already described before in BP patients whereas others are not.

Reply: Thanks for your suggestions. We have added the abbreviations in Supplementary data 15.

The Figure is designed to illustrate the enriched signaling pathways in BP samples. Among the 15 BP-specifically enriched pathways identified, SPP1 signaling emerged as the top pathway. However, it's worth noting that SPP1 signaling originates from Langerhans cells and targets all cell clusters, lacking specific targeted cell types. Conversely, the second most significant signaling pathway, IL4 (a classical pathway for type 2 inflammation), originates from Th2 cells and specifically targets six major subsets (as depicted in **Reviewer Figure 6**).

Reviewer Figure 6. Circle plot of the inferred SPP1 and IL4 signaling pathways among major cell types in the BP group.

Furthermore, within the IL4 pathway, one of the prominent ligand-receptor pairs is IL13/IL13RA1 (**Figure 3f**), the significance of which remains largely unknown in the pathogenesis of dermatological diseases. Therefore, our subsequent studies are focused on investigating the IL4 pathway in greater detail.

7. Figure 4f. In the skin staining PLA2G2A seems to be exclusively cytosolic, which is not in line with the later interpretations. In addition, it would be nice to see a double staining with the respective fibroblast markers to confirm the c. Adding the DEJ by lines or arrows would be helpful to understand the skin orientation. “Immunofluorescence analysis showed that PLA2G2A⁺ fibroblasts had a similar spatial distribution to CD68⁺ macrophages and CD3⁺ T cells ” – For me it looks rather that macrophages concentrate near the DEJ (to the left?) whereas T cell are located in lower layers?

Reply: Thanks for your suggestions. We have done it accordingly.

(1) We conducted a thorough examination of the location of PLA2G2A and confirmed that it is exclusively expressed in the cytoplasm. Our investigation was corroborated by findings from the Human Protein Atlas (<https://www.proteinatlas.org/ENSG00000188257-PLA2G2A>), which categorizes PLA2G2A as an intracellular and cytoplasmic protein. Moreover, recent publications have also consistently reported the cytoplasmic localization of PLA2G2A (Nat Commun. 2022;13(1):6823; BMC Biol. 2022;20(1):276).

Although PLA2G2A is a secreted protein, it is difficult to observe the expression outside the cells by IF staining. Thus, in Figure 4f, PLA2G2A appears to be exclusively cytosolic.

(2) We presented the result of the double staining (**Reviewer Figure 7, new Supplementary Fig. 4a**) using the fibroblast marker PDGFRA (Nature. 2022; 601(7891):118-124; Sci Adv. 2023; 9(4): eadd8977) and PLA2G2A, and the results was consistent with the scRNA-seq data.

Reviewer Figure 7. The IF staining of PLA2G2A/PDGFR α in control and BP skin samples.

(3) Based on the immunofluorescence staining results in **Figure 4f** and the immunohistochemistry (IHC) findings in **Figure S1e**, macrophages indeed tend to aggregate near the dermal-epidermal junction (DEJ), primarily beneath or in proximity to the bullae, while T cells are predominantly positioned in deeper layers. The PLA2G2A⁺ fibroblasts, as depicted in **Reviewer Figure 8 (Figure 4f)**, are situated between CD68⁺ macrophages and CD3⁺ T cells, suggesting their potential roles in immune cell recruitment.

Reviewer Figure 8. Representative images of BP patients and HCs stained by multicolored IHC; green represents PLA2G2A⁺ fibroblasts, red represents macrophages, and the yellow represents CD3 T cells.

Reference:

1. Liu T, Liu C, Yan M, Zhang L, Zhang J, Xiao M, Li Z, Wei X, Zhang H. Single cell profiling of primary and paired metastatic lymph node tumors in breast cancer patients. *Nat Commun.* 2022 Nov 10;13(1):6823. doi: 10.1038/s41467-022-34581-2. PMID: 36357424; PMCID: PMC9649678.
2. Shi JW, Lai ZZ, Yang HL, Zhou WJ, Zhao XY, Xie F, Liu SP, Chen WD, Zhang T, Ye JF, Zhou XY, Li MQ. An IGF1-expressing endometrial stromal cell population is associated with human decidualization. *BMC Biol.* 2022 Dec 8;20(1):276. doi: 10.1186/s12915-022-01483-0. PMID: 36482461; PMCID: PMC9733393.

3. Xu Z, Chen D, Hu Y, Jiang K, Huang H, Du Y, Wu W, Wang J, Sui J, Wang W, Zhang L, Li S, Li C, Yang Y, Chang J, Chen T. Anatomically distinct fibroblast subsets determine skin autoimmune patterns. *Nature*. 2022 Jan;601(7891):118-124. doi: 10.1038/s41586-021-04221-8. Epub 2021 Dec 15. PMID: 34912121.

4. Liu C, Zhang M, Yan X, Ni Y, Gong Y, Wang C, Zhang X, Wan L, Yang H, Ge C, Li Y, Zou W, Huang R, Li X, Sun B, Liu B, Yue J, Yu J. Single-cell dissection of cellular and molecular features underlying human cervical squamous cell carcinoma initiation and progression. *Sci Adv*. 2023 Jan 27;9(4):eadd8977. doi: 10.1126/sciadv.add8977. Epub 2023 Jan 27. PMID: 36706185; PMCID: PMC9882988.

8. Figure S4. For me the further investigation of the CXCL12 / CXCR4 pathway is somehow unclear. Looking to the analysis, i.e., MIF seems to be more interesting. What was the reason to focus on CXCL12 / CXCR4? Again, IF stainings would be interesting for conformation of the proposed hypothesis.

Reply: Thanks for your suggestions.

(1) **Figure S4** illustrates the heightened pathway activity in 0_CCL19+FB and 1_WIF1+FB clusters. While we acknowledge that the MIF/CD74 pathway exhibits a stronger signal compared to the CXCL12/CXCR4 pathway, our primary aim was to investigate the mechanism of immune infiltration mediated by fibroblasts in this study. MIF is extensively expressed across various cell types, whereas CXCL12 is specifically expressed in three types of stromal cells, particularly fibroblasts (as demonstrated in **Reviewer Figure 9**). Furthermore, the MIF/CD74 pathway exhibits significance across all cell clusters, whereas the CXCL12/CXCR4 pathway is particularly enriched in fibroblast sub-populations. Hence, we opted to delve deeper into the CXCL12/CXCR4 pathway for further investigations.

Reviewer Figure 9. The feature plots showing *MIF*, *CXCL12*, *CD74* and *CXCR4* expression within all skin cells.

(2) Following your guidance, we conducted immunofluorescence stainings of PDGFRA/CXCL12, CD3/CXCR4, and CD68/CXCR4. The outcomes revealed that CXCL12 is expressed in PDGFRA+ fibroblasts, while CXCR4 is co-localized in CD3+ T cells and CD68+ macrophages (as depicted in **Reviewer Figure 10**,

Supplementary Fig. 6c). These findings suggest that fibroblasts in BP patients facilitate the infiltration of CXCR4+ T cells and macrophages through the secretion of CXCL12.

Reviewer Figure 10. The IF staining of PDGFRA/CXCL12, CD3/CXCR4 and CD68/CXCR4 in control and BP skin samples.

9. Figures 4i and j. The data seem a bit overinterpreted to me as there is no obvious difference between BP and controls. In figure 4i, the difference is probably given by the fact that PLA2G2A is expressed in BP only? Again, double staining would be interesting to confirm the scRNA data.

Reply: Thanks for your suggestions. We concur that the interpretation of the data in **Figure 4i** and **j** may have been somewhat exaggerated. The enhanced correlation observed in BP lesions primarily stemmed from the up-regulation of PLA2G2A. We have revised in the manuscript as “Due to the up-regulation of *PLA2G2A*, the correlation was much stronger in BP lesions, comparing to normal samples” (Line 255 - 256).

We have done the double staining of PLA2G2A/CXCL12 accordingly. The results showed that PLA2G2A and CXCL12 were co-expressed in BP skin lesions (**Reviewer Figure 11, new Supplementary Fig. 6d**).

Reviewer Figure 11. The IF staining of PLA2G2A/CXCL12 in control and BP skin samples.

10. Figure 5. Similarly to figure 4, the most differences in ELISAs are given by 3 patients that show an effect whereas all other patients do not? Did you check for BP230 level, age and sex here? Maybe this is just a sub-cohort? In addition, in figure 5j/k no effect for HC is expected as the autoreactive cells are missing.

Reply: Thanks for your suggestion.

(1) In addition to the three patients, the remaining 21 samples also displayed a trend of increasing albeit with relatively minor differences. To highlight the disparity between pre- and post-stimulation (PLA2G2A or CCL17), we illustrated the values of PLA2G2A minus control or CCL17 minus control in the right panel of each subfigure (**Reviewer Figure 12**). For the majority of samples, the values are predominantly >0 , indicating that the titer of anti-BP180 NC-16A increased in most samples following stimulation.

Reviewer Figure 12. The results in validation cohort 3 including 53 BP samples with 29 control samples. (a,b) IL-13 secretion (c,d) anti-BP180-NC16A antibody production (e,f) anti-BP230 antibody production, (g) CCL17 secretion.

Following your advice, we examined the BP230 level, age, and sex of the three samples exhibiting significant changes after stimulation (**Reviewer Table 3**). Upon inspection of **Reviewer Table 4**, there is no distinct specificity or difference in the BP230 level, age, and sex compared to the other samples.

Reviewer Table 3. The BP230 level, age and sex of the three samples with great changes after stimulation.

No	Gender	Age	Anti-BP230
1	Female	45	0
2	Male	79	80.9
3	Male	71	0

Reviewer Table 4. The compared results of the BP230 level, age and sex between the three patients (group 1) and the remaining patients (group 2).

Characteristics	Group		P values	
	Group 1	Group 2		
BP230, n	positive	1	4	0.52 (Fisher's exact test)
	negative	2	17	
Age, y (mean±SD)	65.00±17.78	61.81±14.58		0.76 (unpaired two-sided Student's t-test)
Sex, n	male	2	13	1.00 (Fisher's exact test)
	female	1	8	

(3) Yes, given the deficiency of autoreactive cells, the HC samples are not able to produce autoantibodies.

11. Figure 5g. “These results indicated that the fibroblast-derived PLA2G2A may drive the secretion of CCL17 from myeloid cells, especially the DC clusters, in BP patients.” This seems speculative. To confirm, you should include in vitro experiments. Overall, most (not all) interpretations focus on RNA data or ELISA from serum. To confirm the proposed mechanism, the respective cell types (f.e. fibroblasts, myeloid cells) could be stimulated in vitro and the target proteins could be detected on protein levels by WB, IF or FACS. If not possible, this limitation of the study has to be clearly discussed.

Reply: Thanks for your suggestions. We have conducted the new in vitro experiments as per your instructions.

The respective cell types were sorted by FACS and stimulated in vitro to examine the target proteins. However, due to the nature of CCL17 and PLA2G2A being secreted into the supernatant and not being located within the respective cells after stimulation, it becomes challenging to measure the levels of these target proteins using WB, IF, or FACS. Given that ELISA is a standard method for testing supernatant samples, we opted to perform these experiments using ELISA.

DCs were sorted by FACS from both BP patients and controls, and subsequently stimulated with PLA2G2A in vitro for seven days. The secretion of CCL17 in the supernatant was then measured. The findings revealed that DCs exhibited an elevated level of CCL17 production after PLA2G2A stimulation (**Reviewer Figure 13a, Supplementary Fig. 8c**).

Primary fibroblasts derived from human foreskin were stimulated with IL-13 recombinant protein, and the secretion of PLA2G2A in the supernatant was assessed. The results demonstrated an increase in the production of PLA2G2A by fibroblasts following IL-13 stimulation (**Reviewer Figure 13b**).

Reviewer Figure 13. The CCL17 production from sorted DCs (a) and the PLA2G2A secretion from primary fibroblasts (b).

12. Figure 5j/k. The effect of CCL17 and PLA2G2A on autoantibody production is very interesting. However, no B cells are found in the clusters in the skin (figure 1B). Therefore, another effect seems to be important in the skin. Can you comment on this? (Just a comment: Furthermore, no clusters for granulocytes and mast cells are visible in figure 1B - which I know is technically very challenging. However, all interpretation made by lesional scRNA data here miss the effect of these important innate immune cells).

Reply: Thanks for your comments and your understanding of the technical challenges.

Although B cells do exist in the skin of BP patients, their presence was not observed in our lesional scRNA-seq data due to limitations such as the restricted number of cells sequenced and the low frequency of B cells. We acknowledge this limitation, as we also failed to identify granulocytes in our data. In an in vitro environment, granulocytes are highly sensitive to conditions such as temperature, which can result in the rapid loss of cellular activity. Therefore, no cluster for granulocytes is visible in Figure 1b.

However, we did observe a cluster containing mast cells and plasmacytoid dendritic cells (pDC) in **Figure 1b**.

To preliminarily assess the involvement of innate immune cells in BP, we conducted immunohistochemistry staining for CD19 (to define B cells), MPO (to identify neutrophils), and Tryptase (to detect mast cells). The results revealed a significant increase in the numbers of these cells in BP patients compared to controls (**Reviewer Figure 14, new Supplementary Fig. 15**).

Based on our data, we indeed have not conducted research on innate immune cells. In future work, we can continue to explore the role of innate immune cells in BP. We have discussed the aforementioned limitation in the discussion section: “We acknowledge the limitations of our study. Firstly, in skin scRNA data, clusters for B cells and granulocytes were not observed, and only a small cluster of mast cells was identified. Subsequent work will further explore the role of innate immune cells in BP.” (Line 446 - 448)

Reviewer Figure 14. The CD19, MPO and Tryptase stainings.

13. With figure 7, the authors provide a scheme how they envision the interaction circle in BP. To the reviewer this scheme is somewhat misleading, as from the current depiction, the release of PLA2G2A and CXCL12 seems to trigger the initial response. Furthermore, (i) focus only on IL-13 (not IL-4) is somehow over-interpret (ii) what B cells / plasma cells are known to be in the skin in BP patients? (iii) is PLA2G2A really secreted by skin fibroblasts (see figure 3f)? (iv) according to figure 3f the source of IL13 is Th2 and pDC/Mast and proliferating T cells?

Reply: Thank you very much for your constructive suggestion. Based on your feedback and inquiries, we have made the following modifications and provided responses as follows:

We have revised the interpretation of this scheme in the Figure legends (Line 859 - 870). It illustrates a positive feedback loop where fibroblasts respond to Th2 cells

through the IL13RA1-IL13 pair, leading to increased secretion of PLA2G2A and CXCL12. This, in turn, amplifies the Th2-mediated response.

(i) Both IL-13 and IL-4 are crucial type 2 inflammatory cytokines. However, we prioritized IL-13 over IL-4 for the following reasons: 1) The expression level of IL-13 is significantly higher than that of IL-4 (**Reviewer Figure 15**). 2) IL13-IL13RA1 emerges as the most significant ligand-receptor pair within IL4 signaling in BP samples, while IL-4 related signals were not observed in CellChat results. 3) Additionally, IL-13 appears to play a more prominent role in type 2 inflammation-driven skin conditions (Allergy. 2020;75(1):54-62). Hence, our focus in this study was on the role of IL-13 rather than IL-4.

(ii) Previous publications have reported the presence of CXCR4+B cells and CXCR4+ plasma cells in the skin of BP patients (J Invest Dermatol. 2023;143(2):197-208.e6), and IL-13 has been shown to promote B cell maturation into plasma cells (Front Immunol. 2022; 13:824110). Although the B cell cluster was not observed in our skin single-cell RNA sequencing data, immunostaining of CD19 and CD138 confirmed the presence of B cells and plasma cells in BP lesions (**Reviewer Figure 15**).

(iii) Yes, skin fibroblasts secrete PLA2G2A. The primary fibroblasts derived from human foreskin were stimulated by IL-13 recombinant protein and the production of PLA2G2A was increased, indicating that PLA2G2A is really secreted by skin fibroblasts (**Reviewer Figure 13b**).

(iv) According to figure 3f and the lesional scRNA data, the source of IL13 is Th2 and pDC/Mast and proliferating T cells.

Reviewer Figure 15. The expression of IL13 and IL4 in immune cells (a), the CD19 (b) and CD138 (c) stainings to define B cells and plasma cells.

Reference:

1. Bieber T. Interleukin-13: Targeting an underestimated cytokine in atopic dermatitis. *Allergy*. 2020 Jan;75(1):54-62. doi: 10.1111/all.13954. Epub 2019 Jul 15. PMID: 31230370.
2. Fang H, Xue K, Cao T, Li Q, Dang E, Liu Y, Zhang J, Qiao P, Chen J, Ma J, Shen S, Pang B, Bai Y, Qiao H, Shao S, Wang G. CXCL12/CXCR4 Axis Drives the Chemotaxis and Differentiation of B Cells in Bullous Pemphigoid. *J Invest Dermatol*. 2023 Feb;143(2):197-208.e6. doi: 10.1016/j.jid.2022.08.041. Epub 2022 Sep 6. PMID: 36075452.
3. Wang Y, Mao X, Liu Y, Yang Y, Jin H, Li L. IL-13 Genetic Susceptibility to Bullous Pemphigoid: A Potential Target for Treatment and a Prognostic Marker. *Front Immunol*. 2022 Jan 24;13:824110. doi: 10.3389/fimmu.2022.824110. PMID: 35140724; PMCID: PMC8818855.

Minor comments: The authors tend to switch between stromal cells and fibroblasts, which to my understanding they use synonymously. The manuscript is overall understandable when reading carefully. However, the text would benefit from professional revision regarding formulation and grammar, easing up longer paragraphs and choice of appropriate, scientific wording. Some terms like IL4/IL-4 are not used consistently. Also the writings for RNA and proteins are not consistent.

Reply: Thanks for your suggestions. We have corrected some inappropriate descriptions in the manuscript.

Reviewer #2 skin inflammation (Remarks to the Author):

This study performed the scRNA-seq using lesional skin samples, PBMCs, and blister fluids of BP and demonstrates novel findings regarding the crosstalk between fibroblasts and immune cells, that may lead to a better understanding of the pathogenesis of BP and the development of new therapies. I have the following comments which may be helpful for the improvement of the work.

1) What kind of lesions were obtained for scRNA-seq analysis, blister or erythema? It should affect the characteristics of fibroblasts and immune cells. Please clarify it.

Reply: Thanks for your comment. We collected both blister samples and the surrounding erythema for scRNA-seq. We have provided a detailed description of sample collection in the Methods section “Both blister samples and the surrounding erythema were collected for skin scRNA-seq.” [line 468]

2) Langerhans cells often play an important role in the immune response in the skin. Fig1e shows a lower percentage of Langerhans cells in BPs compared to HCs. On the other hand, Fig3e shows that IL13-IL13RA1 is very high in Langerhans cells. It is better to mention the significance of Langerhans cells in the discussion.

Reply: Thanks for your suggestions. In our single-cell RNA sequencing data, we observed an increasing trend in the absolute number of Langerhans cells (**Reviewer Figure 16**). Additionally, immunohistochemistry staining revealed an increased number of Langerhans cells (CD207+) in BP patients (**Reviewer Figure 17a, Supplementary Fig. 2g**), consistent with previous publications (Acta Derm Venereol. 1987;67(6): 529-32). However, the proportion of Langerhans cells among myeloid cells in BP was decreased compared to normal controls due to the marked elevation of dendritic cells in patients.

Reviewer Figure 16. The absolute number of Langerhans cells in the skin RNA-seq data.

Figure 3e illustrates that IL13-IL13RA1 interaction is notably high in Langerhans cells, primarily attributed to the elevated expression of IL13RA1 (**Reviewer Figure 17b**). The cell-cell communication data indicates that Langerhans cells are responsive to IL-13, underscoring their significance in type 2 inflammation in BP patients.

We have included the significance of Langerhans cells in the result part of the revised manuscript:” Due to the marked elevation of DCs, the proportion of Langerhans cells among myeloid cells in BP decreased compared to normal controls. Actually, an increase in the number of Langerhans cells was observed via immunohistochemistry staining in BP patients, which aligns with findings from previous publications³⁴, indicating the significance of Langerhans cells in BP pathogenesis.” [line 132 - 136].

Reviewer Figure 17. The Langerhans cells in control and BP skin samples (a), the plot showing *IL13RA1* expression in immune cells (b).

Reference:

1. Emtestam L, Hovmark A, Lindberg M, Asbrink E. Human epidermal Langerhans' cells in bullous pemphigoid. Acta Derm Venereol. 1987;67(6):529-32. PMID: 2451382.

3) Please explain in detail how to read the hierarchical plot showing inferred intercellular communication network (Fig 3f, 4h, and 5a) in the figure legends or method section, e.g., what do the two opposing arrows at the top of the figure mean?

Reply: We apologize for the lack of detailed explanation in the original text. We have provided a comprehensive explanation using IL13/IL13RA1 as an example in **Reviewer Figure 18 (Figure 3f)**.

In the left part: The six T cell clusters in the middle serve as the signal-receiving cells. The left half of the diagram illustrates autocrine signals, where signals released by the six T cell clusters act on themselves. Correspondingly, the right half shows paracrine signals, where signals released by other types of cells act on the six T cell clusters.

In the right part: The non-T cell clusters serve as the signal-receiving cells. The left half of the diagram represents paracrine signals, where signals released by the six T cell clusters act on these non-T cell clusters. Conversely, the right half depicts autocrine signals, where signals released by these non-T cell clusters act on themselves.

Hence, the two opposing arrows at the top represent the autocrine or paracrine signals received by the middle clusters originating from left or right clusters.

According to your suggestion, we have explained the hierarchical plot in Figure legends (**Figure 3f**).

Reviewer Figure 18. Hierarchical plot showing inferred intercellular communication network of IL13-IL13RA1 signaling in BP skin. Left and right portions show autocrine and paracrine signaling, respectively.

4) Fig 4e shows the function of PLA2G2A on monocytes using THP1, but it is better to also look at the function of PLA2G2A on T cells.

Reply: Thanks for your suggestions. We have done it accordingly. The Jurkat T cells were used to test the recruitment of PLA2G2A on T cells. We treated Jurkat T cells with PLA2G2A recombinant protein and found that PLA2G2A promoted the migration of Jurkat T cells (**Reviewer Figure 19, new Fig. 4e**). This result has been added in Figure 4e in the new manuscript.

Reviewer Figure 19. Transwell assays were used to measure cell migration of Jurkat T cells treated by PLA2G2A recombination protein.

5) The results in Fig 4d are interesting. Please show the correlation between serum PLA2G2A level and BPDAI.

Reply: Thanks for your suggestions. We are sorry that we failed to obtain the records of BPDAI for BP patients from the medical records. Thus, we used BSA scores and the maxim dosage of corticosteroids for BP treatment as an alternative, which could also reflect the severity of the disease. Both the results showed positive correlation between serum PLA2G2A and BSA scores and serum PLA2G2A and the maximum dosage of corticosteroids, respectively (**Reviewer Figure 20, new Supplementary Fig. 4b, c**). These results indicated that serum PLA2G2A could potentially represent a promising marker to define the severity of the disease.

Reviewer Figure 20. The positive correlation between the level of serum PLA2G2A and BSA score (n = 42), between serum PLA2G2A and maximum dosage of corticosteroids (n = 36) in BP patients.

6) Regarding Fig 5g, h, i, it is unclear which cells PLA2G2A or CCL17 worked on since you are using PBMC. You should do the experiment with PBMCs from which DCs and/or T cells were removed to confirm that they are really DCs or T cells.

Reply: Thanks for your suggestions.

In our study, we observed that stimulation with PLA2G2A promoted the secretion of CCL17 (**Figure 5g**) and IL-13 (**Figure 5i**), while CCL17 elevated the production of IL-13 (**Figure 5h**). According to the single-cell RNA sequencing data, CCL17 is primarily expressed by DCs, whereas IL-13 is mainly expressed by T cells. Therefore, we hypothesized that PLA2G2A acts on both DCs and T cells, while CCL17 specifically affects T cells.

To confirm this hypothesis, we performed the following experiments.

(1) The DCs were sorted by FACS from BP patients and controls, and were stimulated with PLA2G2A in vitro for seven days. The results showed that the DCs produced elevated level of CCL17 after PLA2G2A stimulation (**Reviewer Figure 21a, new Supplementary Fig. 8c**).

(2) The T cells were sorted by FACS from BP patients and controls, and were stimulated with PLA2G2A or CCL17 in vitro for seven days. The results showed that the T cells produced elevated level of IL-13 after PLA2G2A (**Reviewer Figure 21b, new Supplementary Fig. 8d**) or CCL17 (**Reviewer Figure 21c, new Supplementary Fig. 8e**) stimulation.

Reviewer Figure 21. The CCL17 production from sorted DC cells (a) and the IL-13 secretion from sorted T cells stimulated by PLA2G2A (b) or CCL17 (c).

7) Regarding Fig 5j and k, it is difficult to understand why anti-BP230 antibodies are not elevated by PLA2G2A and CCL17. Only samples from patients who were positive for anti-BP230 antibodies should be re-examined.

Reply: Thanks.

Following your suggestion, we enrolled eight patients with singular BP230 reactivity and re-examined the secretion of anti-BP230 antibody after PLA2G2A or CCL17 stimulation (**Reviewer Figure 22**). However, we did not detect the secretion of anti-BP230 antibody in these patients, despite their sera showing singular BP230 reactivity. We speculate that this could be due to the anti-BP230 antibody not being a crucial pathogenic factor in the pathogenesis of BP (*Br J Dermatol.* 2017; 177(1):141-151; *Asian Pac J Allergy Immunol.* 2021 Dec;39(4):272-278).

Reviewer Figure 22. The anti-BP230 antibody production from BP patients with singular BP230 reactivity (**a, b**).

Reference:

1. Hashimoto T, Ohzono A, Teye K, Numata S, Hiroyasu S, Tsuruta D, Hachiya T, Kuroda K, Hashiguchi M, Kawakami T, Ishii N. Detection of IgE autoantibodies to BP180 and BP230 and their relationship to clinical features in bullous pemphigoid. *Br J Dermatol.* 2017 Jul;177(1):141-151. doi: 10.1111/bjd.15114. Epub 2017 Apr 12. PMID: 27716903.
2. Chanprapaph K, Ounsakul V, Pruettivorawongse D, Thadanipon K. Anti-BP180 and anti-BP230 enzyme-linked immunosorbent assays for diagnosis and disease activity tracking of bullous pemphigoid: A prospective cohort study. *Asian Pac J Allergy Immunol.* 2021 Dec;39(4):272-278. doi: 10.12932/AP-231118-0446. PMID: 31175713.

Reviewer #3 scRNA and systems (Remarks to the Author):

I conducted a thorough examination primarily centered on data and methods, with a specific focus on assessing quality of the analyses and the reproducibility of the procedures outlined in the current manuscript.

Major points:

- 1) While the procedure outlined for single-cell data processing is reasonably clear, to guarantee reproducibility, it is essential for the authors to share the code used for all the bioinformatics steps mentioned in the Method section, including single-cell preparation, RNA sequencing, and data analysis, on a GitHub repository. Moreover, the code used to generate Figure 1b-d, Figure 2a-f, Figure 3a-f, Figure 4a-e,g-j, Figure 5a-k, Figure 6a-g must be provide. Same apply to supplementary figures. The code must be linked to the data used in the analysis, which must be provided as part of the github or as zenodo/figshare repository.

Reply: Thank you for your comments. The code used for the bioinformatics steps and code used to generate Figures and supplementary figures in this study was uploaded to GitHub (https://github.com/zzwang1030/scRNA_BP) and this information was

provided in the Code availability section of the revised manuscript as follow: "Representative code is available on GitHub (https://github.com/zzwang1030/scRNA_BP). This code is openly available with no restriction or time limit. Any queries or further requests can be addressed to the corresponding authors." (line 582 - 584 in the revised manuscript).

2) In Method section Single-cell preparation, RNA sequencing, and data analysis the authors wrote: "Cluster marker genes were identified using the "FindAllMarkers" function with a filter condition of $\log_2(\text{Foldchange}) (\log_2\text{FC}) > 0.25$ and adjusted P values < 0.05 . In the subclustering of immune cells and fibroblasts, potential doublets were identified and subsequently removed from further analysis. Differential gene expression analysis between the cases and controls was performed using the "FindMarkers" function with a filter condition of $|\log_2\text{FC}| > 0.585$ and adjusted P values < 0.05 ."

The authors should provide justification for selecting $\log_2\text{FC}$ values of only 0.25 and 0.585. These values are exceptionally small, and the authors must illustrate that they exceed the background levels in those experiments.

Reply: Thank you for your comments.

To address concerns about background levels, our team has adjusted the threshold for cluster marker genes from $\log_2\text{FC} > 0.25$ to $\log_2\text{FC} > 1$. Similarly, the threshold for differential gene expression analysis between cases and controls has been adjusted from $|\log_2\text{FC}| > 0.585$ to $|\log_2\text{FC}| > 1$.

Furthermore, it's important to note that the annotation of clusters is not solely based on the cluster marker genes identified through the mentioned threshold. We also utilize canonical cell type-defining signature genes from the top $\log_2\text{FC}$ genes in the process. Similarly, when examining differentially expressed genes (DEGs) between case and control groups, we select the top genes for further analysis.

These adjustments have had no impact on our results.

3) I remain unconvinced about the methodology employed in the Cell-Cell Interaction Analysis section. While CellChat serves as a tool for assessing cell-to-cell communication, its application to skin lesions with distinct multi-layer structure does not ensure that the identified cell-to-cell communication is occurring between cells that are truly in contact.

Reply: Thanks for your comment. CellChat is a powerful tool capable of quantitatively inferring and analyzing intercellular communication networks from scRNA-seq data. The cell-to-cell communication is based on the expression of ligands and receptors. Although the results provided by CellChat are inferred from data, they offer strong indications. In an article by the developers of the CellChat software, they evaluated their method using published skin datasets. They concluded that "Applying

CellChat to mouse and human skin datasets shows its ability to extract complex signaling patterns" (Nat Commun. 2021;12(1):1088).

Based on our data, we identified the ligand-receptor pair CCL19-CCR7 from fibroblasts to DCs, which has been reported in Atopic Dermatitis patients, indicating its role in communication from fibroblasts to DCs (Nat Commun. 2021;12(1):1088). To validate the accuracy of the IL13-IL13RA1 pair in the interaction between Th2 cells and fibroblasts, we conducted immunofluorescence staining of CD3/IL13 and FGFR/IL13RA1 (**Reviewer Figure 23**). Our findings revealed that IL13 was colocalized with CD3+ T cells, while IL-13RA1 was expressed on PDGFRA+ fibroblasts, thus confirming the reliability of the results obtained by CellChat.

Reviewer Figure 23. IF staining of CD3/IL13 and FDGFR/IL13RA1 in BP and HC samples.

Reference:

1. Jin S, Guerrero-Juarez CF, Zhang L, Chang I, Ramos R, Kuan CH, Myung P, Plikus MV, Nie Q. Inference and analysis of cell-cell communication using CellChat. Nat Commun. 2021 Feb 17;12(1):1088. doi: 10.1038/s41467-021-21246-9. PMID: 33597522; PMCID: PMC7889871.

4) Cell type annotation must be better described. Specifically is not clear how the different cell types were defined.

Reply: We apologize for any previous confusion regarding cluster annotation. To validate cluster annotation, we overlapped cluster marker genes with canonical signature genes that define cell types, similar to methods used in published studies (Nat Commun. 2023;14(1):3455; Nat Commun. 2024;15(1):945).

In the revised manuscript, we have included a description of the cell type annotation method in the Methods section: "The annotation of clusters were validated by overlaying the cluster marker genes with canonical signature genes that define cell types." (line 518 - 519).

Furthermore, we have provided specific examples of markers used for the identification of cell types in the Results section. For instance, “By overlapping the cluster marker genes with manual curation of canonical markers, nine main cell types: keratinocytes (KC; *KRT1*, *KRT5*, *KRT10*, *KRT14*), fibroblasts (*DCN*, *COL1A1*, *COL1A2*), dendritic cells/macrophages (DC/Mac; *PTPRC*, *CD68*, *CD1C*), T/Nature killer (NK) cells (*PTPRC*, *CD3D*, *GNLY*, *NKG7*).” (Line 91 - 94). Additionally, the dotplot data illustrating specific markers for each cluster can be found in **Supplementary Fig 1c**, **Supplementary Fig 2c, e**, **Supplementary Fig 3b, d**, **Supplementary Fig 11a-c**, and **Fig.6e**. Furthermore, detailed information on specific markers for each cluster is available in Supplementary data 2, 4, 6, 8, 12, 14.

Reference:

1. Ma F, Plazyo O, Billi AC, et.al. Single cell and spatial sequencing define processes by which keratinocytes and fibroblasts amplify inflammatory responses in psoriasis. *Nat Commun.* 2023 Jun 12;14(1):3455. doi: 10.1038/s41467-023-39020-4. PMID: 37308489.
2. Punzon-Jimenez P, Machado-Lopez A, Perez-Moraga R, et.al. Effect of aging on the human myometrium at single-cell resolution. *Nat Commun.* 2024 Jan 31;15(1):945. doi: 10.1038/s41467-024-45143-z. PMID: 38296945.

5) The analysis of the integrated data is not sufficient, the analysis of the cell composition of each sample must be provided. Annotation must be done on each individual sample and eventually overlaid to the integrated data.

Reply: Thank you very much for your suggestion! The analysis of the cell composition of each sample and annotation of each individual sample have been provided in **Supplementary Fig. 1**, **Figure 2**, **Supplementary Fig. 3**, **Supplementary Fig. 11** and **Supplementary Fig. 12** in the new manuscript.

The corresponding relationship between the dataset and the figure/Supplementary figure is shown in the **Reviewer Table 5**.

Reviewer Table 5. The corresponding relationship between the dataset and the figure/Supplementary figure.

scRNA-seq dataset	Figure link to the cell composition of each sample	Figure link to annotation of each sample
All skin cells	Supplementary Fig. 1f	Supplementary Fig. 1e
Immune cells from skin	Supplementary Fig. 2b,d	Supplementary Fig. 2a
FB cells from skin	Fig. 2b	Supplementary Fig. 3a
KC cells from skin	Fig. 2f	Supplementary Fig. 3c
All PBMC cells	Supplementary Fig. 11d	Supplementary Fig. 11e
All Blister cells	Supplementary Fig. 12d	Supplementary Fig. 12c

Using the scRNA-seq of 13 skin samples as an example, the cell composition of each sample is presented as shown in Supplementary Fig. 1f, while the annotation of each

individual sample is exhibited as shown in Supplementary Fig. 1e (Reviewer Figure 24).

Reviewer Figure 24. **a** The cell composition of each sample of 13 skin scRNA-seq samples (Supplementary Fig. 1f in new manuscript). **b** The annotation of each individual sample of 13 skin scRNA-seq samples(Supplementary Fig. 1e in new manuscript).

5) Metadata containing the cell type annotation associated to the corresponding single cell barcode must be provided for all the cells used in this manuscript.

Reply: Thank you for your comments. All the Metadata have been provided in Supplementary data1,3,5,7,11,13 in the new manuscript. The corresponding relationships are as **Reviewer Table 6**.

Reviewer Table 6. The corresponding relationships between Supplementary Data number and the Metadata.

Number	Content
Supplementary Data 1	The Excel file includes Metadata of all cells from 13 scRNA-seq skin samples.
Supplementary Data 3	The Excel file comprises Metadata pertaining to the subclusters of all immune cells derived from 13 scRNA-seq skin samples.
Supplementary Data 5	The Excel file comprises Metadata pertaining to the subclusters of all fibroblasts cells derived from 13 scRNA-seq skin samples.
Supplementary Data 7	The Excel file comprises Metadata pertaining to the subclusters of all Keratinocytes cells derived from 13 scRNA-seq skin samples.
Supplementary Data 11	The Excel file includes Metadata of all cells from 16 scRNA-seq PBMC samples.
Supplementary Data	The Excel file includes Metadata of all cells from 4

6) Sparse count tables in 10XGenomics format (barcodes.tsv.gz features.tsv.gz matrix.mtx.gz) must also be provided in a figshare/zenodo repository.

Reply: Thank you for your comments. We have uploaded the Sparse count tables in 10XGenomics format of 33 samples to the zenodo website, and the link is [<https://zenodo.org/records/10924853>] and this information was provided in the Data availability section of the revised manuscript as follow: “The scRNA-seq data of 33 samples in 10X Genomics format are available at zenodo (<https://zenodo.org/records/10924853>).” (line 577 - 578 in the revised manuscript).

As this article has not yet been accepted, the data in Zenodo is currently marked as "Restricted." After the article is accepted and published, our data will be made publicly available.

We have provided an exclusive link for reviewer to review our data, and the link is as follows:

<https://zenodo.org/records/10924853?token=eyJhbGciOiJIUzUxMiIsImIhdCI6MTcxNDE4ODQ3MywiZXhwIjoxNzE5NzIxOTk5fQ.eyJpZCI6ImFIM2E4NDc5LTZyZGZEtNDkzZi1hZWQ2LTUzMzY3MzVkMjA3NSIsImRhdGEiOnt9LCJyYW5kb20iOiJlOWVzMzdiMGU0MmRhNzFkNWl4NDdhZGNmZWUxZTcxNSJ9.9FDplWYrSmtDuMFFAqbTXkFe7IXdKvjL6nmae89tAORQBykWXrK0JznVoUfacXk4ZHn6CJXMDMkUvGb0Ozee0g>

7) The overall quality of each single cell RNAseq must be discussed in supplementary data, e.g. using the mitoRiboUmi function available as part of the rCASC package (<https://kedomaniac.github.io/rCASC/reference/mitoRiboUmi.html>)

Reply: The mitoRiboUmi function within the rCASC package has proven to be invaluable. In our current paper, we have utilized both the Seurat (version 4.0.0) R package and the mitoRiboUmi function from the rCASC package to perform rigorous quality control for each single-cell RNAseq sample. This comprehensive approach ensures the precision and reliability of our data.

We added the quality of the 33 single-cell RNA-seq samples in **Supplementary Fig. 1a**, **Supplementary Fig. 10a**, and **Supplementary Fig. 10b** in the new manuscript (**Reviewer Figure 25**).

Reviewer Figure 25. **a** The quality of skin scRNA-seq samples comparing 5BP with 8HC (**Supplementary Fig. 1a** in new manuscript). **b** The quality of PBMC scRNA-seq samples comparing 8BP with 8HC (**Supplementary Fig. 10a** in new manuscript). **c** The quality of blister scRNA-seq samples from 4BP. (**Supplementary Fig. 10b** in new manuscript).

REVIEWERS' COMMENTS

Reviewer #1 (Remarks to the Author):

I have no further comments. The authors addressed all questions and added several new experiments that confirmed their scRNA observation. Congratulation to this nice work.

Reviewer #2 (Remarks to the Author):

The author addressed all my questions adequately.

Reviewer #3 (Remarks to the Author):

The authors have effectively addressed all the points I raised in my review.